**Soil moisture control on sap-flow response to biophysical factors in a desert-shrub**
**species,** *Artemisia ordosica*
**Authors:** Tianshan Zha[1,3*#], Duo Qian[2#], Xin Jia[1,3], Yujie Bai [1], Yun Tian[1], Charles P.-A.
Bourque[4], Wei Feng[1], Bin Wu[1], Heli Peltola[5]
[1.] Yanchi Research Station, School of Soil and Water Conservation, Beijing Forestry
University, Beijing 100083, China
[2.] Beijing Vocational College of Agriculture, Beijing 102442, China
[3.] Key Laboratory of State Forestry Administration on Soil and Water Conservation,
Beijing Forestry University, Beijing, China
[4.] Faculty of Forestry and Environmental Management, 28 Dineen Drive, PO Box 4400,
University of New Brunswick, New Brunswick, E3B5A3, Canada
[5.] Faculty of Science and Forestry, School of Forest Sciences, University of Eastern
Finland, Joensuu, FI-80101, Finland
[#]These authors contributed equally to this work.
**Short title:  Sap flow in** *Artemisia ordosica*
*Correspondence to*: T. Zha (tianshanzha@bjfu.edu.cn),

**Author Contribution Statement:**

Dr.'s Duo Qian and Tianshan Zha contributed equally to the design and implementation of the field experiment, data collection and analysis, and writing the first draft of the manuscript.

Dr. Xin Jia gave helpful suggestions concerning the analysis of the field data and contributed to the scientific revision and editing of the manuscript.

Prof. Bin Wu contributed to the design of the experiment.

Dr.'s Charles P.-A. Bourque and Heli Peltola contributed to the scientific revision and editing of the manuscript.

Yujie Bai, Wei Feng, and Yun Tian were involved in the implementation of the experiment and in the revision of the manuscript.

**Key Message:** This study provides a significant contribution to the understanding of acclimation processes in desert-shrub species to drought-associated stress in dryland ecosystems

**Conflict of Interest:**

This research was financially supported by grants from the National Natural Science Foundation of China (NSFC No. 31670710), the National Basic Research Program of China (Grant No. 2013CB429901), and by the Academy of Finland (Project No. 14921). The project is related to the Finnish-Chinese collaborative research project, EXTREME (2013-2016), between Beijing Forestry University and the University of Eastern Finland, and USCCC. We appreciate Dr. Ben Wang, Sijing Li, Qiang Yang, and others for their help with the fieldwork. **The authors declare that they have no conflict of interest.**

**Abstract:** Current understanding of acclimation processes in desert-shrub species to drought stress in dryland ecosystems is still incomplete. In this study, we measured sap flow in *Artemisia ordosica* and associated environmental variables throughout the growing seasons of 2013-2014 (May-September period of each year) to better understand the environmental controls on the temporal dynamics of sap flow. We found that the occurrence of drought in the dry year of 2013 during the leaf-expansion and leaf-expanded periods caused sap flow per leaf area ($J_s$) to decline significantly, resulting in a sizable drop in transpiration. Sap flow per leaf area correlated positively with radiation ($R_s$), air temperature ($T$), and vapor pressure deficit (VPD), when volumetric soil water content (VWC) was $> 0.10$ $m^3\,m^{-3}$. Diurnal $J_s$ was generally ahead of $R_s$ by as much as 6 hours. This lag time, however, decreased with increasing VWC. Relative response of $J_s$ to the environmental variables (i.e., $R_s$, $T$, and VPD) varied with VWC, $J_s$ being more biologically-controlled with a low decoupling coefficient and less sensitivity to the environmental variables during periods of dryness. According to this study, soil moisture is shown to control sap-flow (and, therefore, plant-transpiration) response in *Artemisia ordosica* to diurnal variations in biophysical factors. The findings of this study add to the knowledge of acclimation processes in desert-shrub species under drought-associated stress. This knowledge is essential to model desert-shrub-ecosystem functioning under changing climatic conditions.

**Keywords:** sap flow; transpiration; cold-desert shrubs; environmental stress; volumetric soil water content

## 1. Introduction

Due to the low amount of precipitation and high potential evapotranspiration in desert ecosystems, low soil water availability limits both plant water- and gas-exchange and, as a consequence, limits vegetation productivity (Razzaghi et al., 2011). Therefore, it is important to understand the mechanisms controlling the vegetation-water dynamics under rapidly changing environments (Jacobsen et al., 2007). Grass species are gradually being replaced by shrub and semi-shrub species in arid and semi-arid areas of northwestern China (Yu et al., 2004). This progression is predicted to continue under a changing climate (Asner et al., 2003; Houghton et al., 1999; Pacala et al., 2001). This is mostly because desert shrubs are able to adapt to hot-dry environments by modifying their morphological characteristics, e.g., by (1) minimizing plant-surface area directly exposed to sun and hot air, (2) producing thick epidermal hairs, (3) thickening cuticle, (4) recessing stomata into leaves (Yang and Zhu, 2011), and (5) increasing root-to-shoot ratios (Eberbach and Burrows, 2006; Forner et al., 2014). Also, acclimation of physiological characteristics of plants under water stress, by way of e.g., water potential, osmotic regulation, anti-oxidation, and photosynthetic characteristics, assist the plants to maintain a hydrological balance (Huang et al., 2011a). Changes in stomatal conductance and, thus, transpiration may likewise affect plant water use efficiency (Pacala et al., 2001; Vilagrosa et al., 2003).

Sap flow can accurately reflect water consumption during plant transpiration. It maintains ecosystem balance through the soil-plant-atmosphere continuum, but is often affected by environment factors (Huang *et al*., 2010; Zhao et al., 2016). In recent studies, sap flow in *Tamarix elongate* has been observed to be controlled by solar radiation and air temperature, whereas in *Caragana korshinskii* vapor pressure deficit and solar radiation

appear to be more important (Jacobsen et al., 2007; Xia et al., 2008). In *Elaeagnus angustifolia*, transpiration is observed to peak at noon, i.e., just before stomatal closure at mid-day under water-deficit conditions (Liu et al., 2011). In contrast, transpiration in *Hedysarum scoparium* peaks multiple times during the day (Xia et al., 2007). Sap flow has been observed to decrease rapidly when the volumetric soil water content (VWC) is lower than the water loss through evapotranspiration (Buzkova et al., 2015). In general, desert shrubs can close their stomata to reduce transpiration when exposed to dehydration stress around mid-day. However, differences exist among shrub species with respect to their stomatal response to changes in soil and air moisture deficits (Pacala et al., 2001). For some shrubs, sap-flow response to precipitation varies from an immediate decline after a heavy rainfall to no observable change after a small rainfall event (Asner et al., 2003; Zheng and Wang, 2014). Sap flow has been found to increase with increasing rainfall intensity (Jian et al., 2016). Drought-insensitive shrubs have relatively strong stomatal regulation and, therefore, tend to be insensitive to soil water deficits and rainfall unlike their drought-sensitive counterparts (Du et al., 2011). In general, understanding of the relationship between sap-flow rates in plants and environmental factors is highly inconsistent, varying with plant habitat (Liu et al., 2011).

*Artemisia ordosica,* a shallow-rooted desert shrub, is the dominant plant species in the Mu Us Desert of northwestern China. The shrubs have an important role in combating desertification and in stabilizing sand dunes (Li et al., 2010). Increases in air temperature and precipitation variability and associated shorter wet periods and longer intervals of periodic drought are expected to ensue with projected climate change (Lioubimtseva and Henebry,

2009). During dry periods of the year, sap flow in *Artemisia ordosica* has been observed to be controlled by VWC at about 30-cm depth in the soil (Li et al., 2014). Sap-flow rate is known to be affected by variation in precipitation patterns. Soil water content, in combination with other environmental factors, may have a significant influence on sap-flow rate (Li et al., 2014; Zheng and Wang, 2014). Thus, understanding the controlling mechanisms of sap flow in desert shrubs as a function of variations in biotic and abiotic factors is greatly needed (Gao et al., 2013; Xu et al., 2007).

In this study, we measured stem sap flow in *Artemisia ordosica* and associated environmental variables throughout the growing seasons of 2013-2014 (May-September period of each year) to better understand the environmental controls on the temporal dynamics of sap flow. We believe that our findings will provide further understanding of acclimation processes in desert-shrub species under stress of dehydration.

## 2. Materials and Methods

### 2.1 Experimental site

Continuous sap-flow measurements were made at the Yanchi Research Station (37°42′ 31″ N, 107°13′ 47″ E, 1530 m above mean sea level), Ningxia, northwestern China. The research station is located between the arid and semi-arid climatic zones along the southern edge of the Mu Us Desert. The sandy soil in the upper 10 cm of the soil profile has a bulk density of 1.54±0.08 g cm$^{-3}$ (mean ± standard deviation, n=16). Mean annual precipitation in the region is about 287 mm, of which 62% falls between July and September. Mean annual potential evapotranspiration and air temperature are about 2,024 mm and 8.1℃ based on

meteorological data (1954-2004) from the Yanchi County weather station. Normally, shrub
leaf-expansion, leaf-expanded, and leaf-coloration stages begin in April, June, and
September (Chen et al., 2015), respectively.

**2.2 Measurements of sap flow, leaf area and stomatal conductance**
The experimental plot (10 m × 10 m) was located on the western side of Yanchi Research
Station in an *Artemisia ordosica*-dominated area. Mean age of the *Artemisia ordosica* was
10-years old. Maximum monthly mean leaf area index (LAI) for plant specimens with full
leaf expansion was about 0.1 $m^2$ $m^{-2}$ (Table 1). Over 60% of their roots were
distributed in soil depths of 0-60 cm (Zhao et al., 2010; Jia et al., 2016). Five stems of
*Artemisia ordosica* were randomly selected within the plot as replicates for sap-flow
measurement. Mean height and sapwood area of sampled shrubs were 84 cm and 0.17 $cm^2$,
respectively. Sampled stems represented the average size of stems in the plot. A heat balance
sensor (Flow32-1K, Dynamax Inc., Houston, USA) was installed at about 15 cm above the
ground surface on each of the five stems (Dynamax, 2005). Sap-flow measurements were
taken once per minute for each stem. Half-hourly data were recorded by a Campbell CR1000
data logger from May 1 to September 30, 2013-2014 (Campbell Scientific, Logan, UT, USA).
Leaf area was estimated for each stem every 7-10 days by sampling about 50-70 leaves
from five randomly sampled neighbouring shrubs with similar characteristics to the shrubs
used for sap-flow measurements. Leaf area was measured immediately at the station
laboratory with a portable leaf-area meter (LI-3000, Li-Cor, Lincoln, NE, USA). Leaf area
index (LAI) was measured at roughly weekly intervals on a 4×4 grid of 16 quadrats (10 m
×10 m each) within a 100 m × 100 m plot centered on the flux tower using measurements of
sampled leaves and allometric equations (Jia et al., 2014). Stomatal conductance ($g_s$) was
measured *in situ* for three to four leaves on each of the sampled shrubs with a LI-6400
portable photosynthesis analyzer (Li-Cor Inc., Lincoln, USA). The $g_s$ measurements were
made every two hours from 7:00 to 19:00 h every ten days from May to September, 2013-

162 2014.

The degree of coupling between the ecosystem surface and the atmospheric boundary
layer was estimated with the decoupling coefficient ($\Omega$). The decoupling coefficient varies
from 0 (i.e., leaf transpiration is mostly controlled by $g_s$) to 1 (i.e., leaf transpiration is mostly
controlled by radiation). The $\Omega$ was calculated as described by Jarvis and McNaughton

167 (1986):

$$\Omega = \frac{\Delta + \gamma}{\Delta + \gamma\left(1 + \dfrac{g_a}{g_s}\right)},\qquad(1)$$
where $\Delta$ is the rate of change of saturation vapor pressure *vs.* temperature (kPa K$^{-1}$), $\gamma$ is the
psychrometric constant (kPa K$^{-1}$), and $g_a$ is the aerodynamic conductance (m s$^{-1}$; Monteith
and Unsworth, 1990):
$$g_a = \left(\frac{u}{u^{*2}} + 6.2u^{*-0.67}\right)^{-1},\qquad(2)$$
where $u$ is the wind speed (m s$^{-1}$) at 6 m above the ground, and $u^*$ is the friction velocity (m
s$^{-1}$).

**2.3 Environmental measurements**
Shortwave radiation ($R_s$ in W m$^{-2}$; CMP3, Kipp & Zonen, Netherland), air temperature ($T$ in
°C), wind speed ($u$ in m s$^{-1}$, 034B, Met One Instruments Inc., USA), and relative humidity
(*RH* in %; HMP155A, Väisälä, Finland) were measured simultaneously near the sap-flow
measurement plot. Half-hourly data were recorded by data logger (CR3000 data logger,
Campbell Scientific Inc., USA). VWC at 30-cm depths were monitored with three ECH$_2$O-
5TE soil moisture probes (Decagon Devices, USA). In the analysis, we used half-hourly
averages of VWC from the three soil moisture probes. Vapor pressure deficit (VPD in kPa)
was calculated from recorded *RH* and *T*.

**2.4 Data analysis**
In our analysis, March-May represented spring, June-August summer, and September-
November autumn (Chen et al., 2015). Drought days were defined as those days with daily
mean VWC < 0.1 m$^3$ m$^{-3}$. This is based on a VWC threshold of 0.1 m$^3$ m$^{-3}$ for $J_s$ (Fig. 1),
with $J_s$ increasing as VWC increased, saturating at VWC of 0.1 m$^3$ m$^{-3}$, and decreasing as
VWC continued to increase. The VWC threshold of 0.1 m$^3$ m$^{-3}$ is equivalent to a relative
extractable soil water (REW) of 0.4 for drought conditions (Granier et al., 1999 and 2007;
Zeppel et al., 2004 and 2008; Fig. 2d, e). Duration and severity of 'drought' were defined
based on a VWC threshold and REW of 0.4. REW was calculated as according to equation

195 (3):

$$REW = \frac{VWC - VWC_{min}}{VWC_{max} - VWC_{min}} \qquad (3)$$

where VWC is the specific daily soil water content (m$^3$ m$^{-3}$), VWC$_{min}$ and VWC$_{max}$ are the
minimum and maximum VWC during the measurement period in each year, respectively.
Sap-flow analysis was conducted using mean data from five sensors. Sap flow per leaf

area ($J_s$) was used in this study, i.e.,

$$J_s = \left( \sum_{i=1}^{n} E_i / A_{li} \right) \Big/ n \qquad (4)$$

where, $J_s$ is the sap flow per leaf area (kg m$^{-2}$ h$^{-1}$) or (kg m$^{-2}$ d$^{-1}$), $E$ is the measured sap flow

of a stem (g h$^{-1}$), $A_l$ is the leaf area of the sap-flow stem, and "$n$" is the number of stems used

($n = 5$).

Transpiration per ground area ($T_r$) was estimated in this study according to:

$$T_r = \left( \sum_{i=1}^{n} J_s \times LAI \right) \Big/ n \qquad (5)$$

where, $T_r$ is transpiration per ground area (mm d$^{-1}$), and LAI is the leaf area index (m$^2$

m$^{-2}$).

Linear and non-linear regression were used to analyze abiotic control on sap-flow rate.

In order to minimize the effects of different phenophases and rainfall, we used data only from

mid-growing season, non-rainy days, and daytime measurements (8:00-20:00), i.e., from

June 1 to August 31, with hourly shortwave radiation > 10 W m$^{-2}$. Relations between mean

sap-flow rates at specific times over a period of 8:00-20:00 and corresponding environmental

factors from June 1 to August 31 were derived with linear regression (p<0.05; Fig. 3).

Regression slopes were used as indicators of sap-flow sensitivity (degree of response) to the

various environmental variables (see e.g., Zha et al., 2013). All statistical analyses were

performed with SPSS v. 17.0 for Windows software (SPSS Inc., USA). Significance level

was set at 0.05.

## 3. Results

### 3.1 Seasonal variations in environmental factors and sap flow

Range of daily means (24-hour mean) for $R_s$, $T$, VPD, and VWC during the 2013 growing season (May-September) were 31.1-364.9 W m$^{-2}$, 8.8-24.4$^\circ$C, 0.05-2.3 kPa, and 0.06-0.17 m$^3$ m$^{-3}$ (Fig. 2a, b, c, d), respectively, annual means being 224.8 W m$^{-2}$, 17.7$^\circ$C, 1.03 kPa, and 0.08 m$^3$ m$^{-3}$. Corresponding range of daily means for 2014 were 31.0-369.9 W m$^{-2}$, 7.1-25.8$^\circ$C, 0.08-2.5 kPa, and 0.06-0.16 m$^3$ m$^{-3}$ (Fig. 2a, b, c, d), respectively, annual means being 234.9 W m$^{-2}$, 17.2$^\circ$C, 1.05 kPa, and 0.09 m$^3$ m$^{-3}$.

Total precipitation and number of rainfall events during the 2013 measurement period (257.2 mm and 46 days) were about 5.6% and 9.8% lower than those during 2014 (272.4 mm and 51 days; Fig. 2d), respectively. In 2013, more irregular rainfall events occurred than in 2014, with 45.2% of rainfall falling in July and 8.8% in August.

Drought mainly occurred in May, June, and August of 2013 and in May and June of 2014 (Fig. 2d,e). Both years had dry springs. Over one-month period of summer drought occurred in 2013.

Range of daily $J_s$ during the growing season was 0.01-4.36 kg m$^{-2}$ d$^{-1}$ in 2013 and 0.01-2.91 kg m$^{-2}$ d$^{-1}$ in 2014 (Fig. 2f), with annual means of 0.89 kg m$^{-2}$ d$^{-1}$ in 2013 and 1.31 kg m$^{-2}$ d$^{-1}$ in 2014. Mean daily $J_s$ over the growing season of 2013 was 32%, lower than that of 2014. Mean daily $T_r$ were 0.05 mm d$^{-1}$ and 0.07 mm d$^{-1}$ over the growing season in 2013 and 2014 (Fig. 2f), respectively, being 34% lower in 2013 than in 2014. The total $T_r$ over growing season (May 1-September 30) in 2013 and 2014 were 7.3 mm and 10.9 mm, respectively. Seasonal fluctuations in $J_s$ and $T_r$ corresponded with the seasonal pattern in VWC (Fig. 2d, f). Daily mean $J_s$ and $T_r$ decreased or remained nearly constant during dry-soil periods (Fig. 2d, f), with the lowest $J_s$ and $T_r$ observed in spring and mid-summer (August) of 2013.

**3.2 Sap flow response to environmental factors**

In summer, $J_s$ increased with increasing VWC (Fig. 2d, f; Fig. 3d).Soil water was shown to modify the response of $J_s$ to environmental factors (Fig. 4). Sap flow increased more rapidly with increases in $R_s$, $T$, and VPD under high VWC (i.e., VWC > 0.1 m$^3$ m$^{-3}$ in both 2013 and 2014) compared with periods with lower VWC (i.e., VWC < 0.1 m$^3$ m$^{-3}$ in both 2013 and 2014). Sap flow $J_s$ was more sensitive to $R_s$, $T$, and VPD under high VWC (Fig. 4), which coincided with a larger regression slope under high VWC conditions.

Sensitivity of $J_s$ to environmental variables (in particular, $R_s$, $T$, VPD, and VWC) varied depending on the time of a day (Fig. 5). Regression slopes for the relations of $J_s$-$R_s$, $J_s$-$T$, and $J_s$-VPD were greater in the morning before 11:00 h, and lower during mid-day and early afternoon (12:00-16:00 h). In contrast, regression slopes of the relation of $J_s$-VWC were lower in the morning (Fig. 5), increasing thereafter, peaking at ~13:00 h, and subsequently decreasing in late afternoon. Regression slopes of the response of $J_s$ to $R_s$, $T$, and VPD in 2014 were greater than those in 2013.

**3.3 Diurnal changes and hysteresis between sap flow and environmental factors**

Diurnal patterns of $J_s$ were similar in both years (Fig. 6), initiating at 7:00 h and increasing thereafter, peaking before noon (12:00 h), and subsequently decreasing thereafter and remaining near zero from 20:00 to 6:00 h. Diurnal changes in $g_s$ were similar to $J_s$, but peaking about 2 and 1 h earlier than $J_s$ in July and August, respectively (Fig. 6).

There were pronounced time lags between $J_s$ and $R_s$ over the two years (Fig. 7), $J_s$ peaking earlier than $R_s$ and, thus, earlier than either VPD or $T$. These time lags differed seasonally. For example, mean time lag between $J_s$ and $R_s$ was 2 h during July, 5 h during

May, and 3 h during June, August, and September of 2013. However, the time lags in 2014
were generally shorter than those observed in 2013 (Table 2).

Use of normalized variables may remove the influence of $J_s$ and $R_s$ from the data. As a

result, clockwise hysteresis loops between $J_s$ and $R_s$ during the growing period were observed
(Fig. 7). As $R_s$ increased in the morning, $J_s$ increased until it peaked at ~10:00 h. Sap-flow
rate declined with decreasing $R_s$ during the afternoon. Sap flow $J_s$ was higher in the morning
than in the afternoon, forming a clockwise hysteresis loop.

Diurnal time lag in the relation of $J_s$-$R_s$ were influenced by VWC (Fig. 8, 9). For

example, $J_s$ peaked about 2 h earlier than $R_s$ on days with low VWC (Fig. 8a), 1 h earlier than
$R_s$ on days with moderate VWC (Fig. 8b), and at the same time as $R_s$ on days with high VWC
(Fig. 8c). Lag hours between $J_s$ and $R_s$ over the growing season were negatively and linearly
related to VWC (Fig. 9: Lag (h) =-133.5×VWC+12.24, $R^2$=0.41). Effect of VWC on time
lags between $J_s$ and $R_s$ was smaller in 2014, with evenly distributed rainfall during the
growing season, than in 2013, with a pronounced summer drought (Fig. 9). State variables $g_s$
and $\Omega$ showed a significantly increasing trend with increasing VWC in 2013 and 2014,
respectively (Fig. 10).

**4. Discussion and conclusions**
**4.1 Sap flow response to environmental factors**
Drought tolerance of some plants may be related to lower overall sensitivity of plant
physiological attributes to environmental stress and/or stomatal regulation (Huang et al.,
2011b; Naithani et al., 2012). In this study, large regression slopes between $J_s$ and the

environmental variables ($R_s$, VPD, and $T$) in the morning indicated that sap flow was more

sensitive to variations in $R_s$, VPD, and $T$ during the less dry and hot period of the day (Fig.

5). Stomatal conductances were the largest in the morning (Fig. 6), which led to increases in

water fluxes to the atmosphere as a result of increased $R_s$, $T$, and VPD. When $R_s$ peaked

during mid-day (13:00-14:00 h), there was often insufficient soil water to meet the

atmospheric demand for water, causing $g_s$ to be limited by available soil moisture and making

$J_s$ more responsive to VWC at noon, but less responsive to $R_s$ and $T$. Similarly, *Hedysarum*

*mongolicum* in a nearby region positively correlated with VWC at noon (Qian et al., 2015),

and the evapotranspiration of a Scots pine stand showed higher sensitivity to surface

conductance, temperature, vapor pressure deficit, and radiation in the morning than in the

afternoon (Zha et al., 2013).

Synergistic interactions among environmental factors influencing sap flow are complex.

In general, VWC has an influence on physiological processes of plants in water-limited

ecosystems (Lei et al., 2010; She et al., 2013). Our finding regarding lower sensitivity in $J_s$

to environmental factors ($R_s$, $T$ and VPD) during dry periods was consistent with an earlier

study of boreal grasslands (Zha et al., 2010). Also our finding that VWC is the most important

factor modifying responses in sap flow in *Artemisia ordosica* to other environmental factors,

is in contrast to other shrub species. For example, it has been found that sap flow in *Haloxylon*

*ammodendron* in northwest China, where annual precipitation is 37.9 mm and mean annual

temperature is 8.2 ℃, was mainly controlled by $T$ (Zhang et al., 2003), while sap flow in

*Cyclobalanopsis glauca* in south China, where annual precipitation is 1900 mm and mean

annual temperature is 19.3 ℃, was controlled by $R_s$ and $T$, when VWC was not limiting

(Huang et al., 2009).
Precipitation, being the main source of VWC at our site, affected transpiration directly.
In this sense, frequent small rainfall events ($<$ 5 mm) were important to the survival and
growth of the desert plants (Sala and Lauenroth, 1982; Zhao and Liu, 2010). Variations in $J_s$
were clearly associated with the intermittent supply of water to the soil during rainfall events,
as indicated at our site (Fig. 2d, f). Reduced $J_s$ during rainy days can be explained by a
reduction in incident $R_s$ and water-induced saturation on the leaf surface, which led to a
decrease in leaf turgor and stomatal closure. After each rainfall event, $J_s$ increased quickly
when soil water was replenished. Schwinning and Sala (2004) showed previously for similar
research sites that VWC contributed the most to the response in plant transpiration to post-
rainfall events. We showed in this study that *Artemisia ordosica* responded in a different way
to wet and dry conditions. In the mid-growing season, high $J_s$ in July were related to rainfall-
fed VWC, which increased the rate of transpiration. However, dry soil conditions combined
with high $T$ and $R_s$, led to a reduction in $J_s$ in August of 2013 (Fig. 2). In some desert shrubs,
groundwater may replenish water lost by transpiration by having deep roots (Yin et al., 2014).
*Artemisia ordosica* roots are generally distributed in the upper 60 cm of the soil (Zhao et al.,
2010; Wang et al., 2016), and as a result the plant usually depends on water directly supplied
by precipitation because groundwater levels in drylands can be well below the rooting zone,
typically, at depths $\geq$ 10 m at our site.

**4.2 Hysteresis between sap flow and environmental factors**
Diurnal patterns in $J_s$ corresponded with those of $R_s$ from sunrise until diverging later in the
day (Fig. 7), suggesting that $R_s$ was a primary controlling factor of diurnal variation in $J_s$.
According to O'Brien et al. (2004), diurnal variation in $R_s$ could cause change in the diurnal
variation in the consumption of water. As an initial energy source, $R_s$ can force $T$ and VPD
to increase, causing a phase difference in time lags among the relations $J_s$-$R_s$, $J_s$-$T$, and $J_s$-
VPD.
We found a consistent clockwise hysteresis loop between $J_s$ and $R_s$ over a diurnal cycle
(Fig. 7), indicating that $R_s$ lagged $J_s$, and the response of $J_s$ to $R_s$ varied both diurnally and
seasonally. A large $g_s$ in the morning promoted higher rates of transpiration (Fig. 6). In dry
and hot conditions, $g_s$ decreased, causing the control of the stomata on $J_s$ to increase relative
to changes in environmental factors. Diurnal trends in $J_s$ and $g_s$ occurred together, both
peaking earlier than $R_s$.. The $g_s$ peaked 3-4 h earlier than $R_s$, leading to a reduction in $J_s$ and
an increase in $R_s$ and a clockwise hysteresis loop. Contrary to our findings, counterclockwise
hysteresis has been observed to occur between transpiration ($J_s$) and $R_s$ in tropical and
temperate forests (Meinzer et al., 1997; O'Brien et al., 2004; Zeppel et al., 2004). A possible
reason for this difference may be due to differences in VWC associated with the different
regions. According to Zheng and Wang (2014) favorable water conditions after rainfall could
render clockwise hysteresis loops between $J_s$ and $R_s$ under dry conditions to counterclockwise
loops. In this study, due to a large incidence of small rainfall events, soil water supply by
rainfall pulses could not meet the transpiration demand under high mid-day $R_s$, resulting in
clockwise loops even though rainfall had occurred.
In semi-arid regions, low VWC restricts plant transpiration more than VPD. Water
vapor deficits tend to restrict transpiration in forest species in wet regions to a greater extent.

According to Zheng et al. (2014), high water availability in alpine shrubland meadows may contribute to weakened hysteresis between evapotranspiration and the environmental variables. Our results showed that hysteresis between $J_s$ and $R_s$ decreased as VWC increased (Fig. 8, 9). The result that $g_s$ increased with increasing VWC (Fig. 10a), along with the synchronization of $J_s$ and $g_s$, suggests that $J_s$ is less sensitive to $g_s$ in high VWC and more so to $R_s$. Temporal patterns in $J_s$ became more consistent with those in $R_s$ as VWC increased, leading to a weakened hysteresis between the two variables. This is further supported by a large decoupling coefficient, when VWC is high (Fig. 10b). The larger the decoupling coefficient is, the greater is the influence of $R_s$ on $J_s$. The effect of VWC on time lag varied between 2013 and 2014.

**4.3. Conclusions**

Drought during the leaf-expansion and leaf-expanded periods led to a greater decline in $J_s$, causing $J_s$ to be lower in 2013 than in 2014. The relative influence of $R_s$, $T$, and VPD on $J_s$ in *Artemisia ordosica* was modified by soil water content, indicating $J_s$'s lower sensitivity to environmental variables ($R_s$, $T$ and VPD) during dry periods. Sap flow $J_s$ was constrained by soil water deficiency, causing $J_s$ to peak several hours prior to $R_s$. Diurnal hysteresis between $J_s$ and $R_s$ varied seasonally, because of the control by stomatal conductance under low VWC and $R_s$ under high VWC. According to this study, soil moisture controlled sap-flow response in *Artemisia ordosica.* This species is capable to tolerate and adapt to soil water deficiencies and drought conditions during the growing season. Altogether, our findings add to our understanding of acclimation in desert-shrub species under stress of dehydration. The knowledge gain can assist in modeling desert-shrub-ecosystem functioning under changing

climatic conditions.

**Acknowledgments:** This research was financially supported by grants from the National Natural Science Foundation of China (NSFC No. 31670710, 31670708, 31361130340, 31270755), the National Basic Research Program of China (Grant No. 2013CB429901), and the Academy of Finland (Project No. 14921). Xin Jia and Wei Feng are also grateful to financial support from the Fundamental Research Funds for the Central Universities (Proj. No. 2015ZCQ-SB-02). This work is related to the Finnish-Chinese collaborative research project EXTREME (2013-2016), between Beijing Forestry University (team led by Prof. Tianshan Zha) and the University of Eastern Finland (team led by Prof. Heli Peltola), and the U.S. China Carbon Consortium (USCCC). We thank Ben Wang, Sijing Li, Qiang Yang, and others for their assistance in the field.

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

 **Table 1** Seasonal changes in monthly transpiration ($T_r$), leaf area index (LAI), and stomatal
 conductance ($g_s$) of *Artemisia ordosica* from 2013 to 2014.

| | $T_r$ (mm mon$^{-1}$) | | LAI (m$^2$ m$^{-2}$) | | $g_s$ (mol m$^{-2}$ s$^{-1}$) | |
|---|---|---|---|---|---|---|
| | 2013 | 2014 | 2013 | 2014 | 2013 | 2014 |
| May | 0.57 | 1.59 | 0.02 | 0.04 | 0.07 | 0.18 |
| June | 1.03 | 2.28 | 0.05 | 0.06 | 0.08 | 0.13 |
| July | 3.36 | 3.46 | 0.10 | 0.06 | 0.09 | 0.14 |
| August | 1.04 | 2.45 | 0.08 | 0.06 | 0.10 | 0.08 |
| September | 1.23 | 1.13 | 0.05 | 0.04 | 0.15 | 0.05 |

**Table 2** Mean monthly diurnal cycles of sap-flow rate ($J_s$) response to shortwave radiation
($R_s$), air temperature ($T$), and vapor pressure deficit (VPD), including time lags (h) in $J_s$ as a
function of $R_s$, $T$, and VPD.

| Pattern | May | | June | | July | | August | | September | |
|---|---|---|---|---|---|---|---|---|---|---|
| | 2013 | 2014 | 2013 | 2014 | 2013 | 2014 | 2013 | 2014 | 2013 | 2014 |
| $J_s$-$R_s$ | 5 | 2 | 3 | 0 | 2 | 1 | 3 | 1 | 3 | 2 |
| $J_s$-$T$ | 8 | 6 | 7 | 4 | 4 | 4 | 6 | 5 | 6 | 6 |
| $J_s$-VPD | 8 | 5 | 7 | 4 | 6 | 4 | 6 | 5 | 6 | 5 |




**Figure captions:**

**Fig. 1** Sap-flow rate per leaf area ($J_s$) as a function of soil water content (VWC) at 30 cm depth in non-rainy, daytime hours during the mid-growing period from June 1-August 31 over 2013-2014. Data points are binned values from pooled data over two years at a VWC increment of 0.003 $m^3\,m^{-3}$. Dotted line represents the VWC threshold for $J_s$.

**Fig. 2** Seasonal changes in daily (24-hour) mean shortwave radiation ($R_s$; a), air temperature ($T$; b), vapor pressure deficit (VPD; c), volumetric soil water content (VWC; d), relative extractable water (REW; e), daily total precipitation (PPT; d), and daily sap-flow per leaf area ($J_s$; f), and daily transpiration ($T_r$, mm $d^{-1}$; f) from May to September for both 2013 and 2014. Horizontal dash lines (d, e) represent VWC and REW threshold of 0.1 $m^3\,m^{-3}$ and 0.4, respectively. Shaded bands indicate periods of drought.

**Fig. 3** Relationships between sap-flow rate per leaf area ($J_s$) and environmental factors [shortwave radiation ($R_s$), air temperature ($T$), vapor pressure deficit (VPD), and soil water content at 30-cm depth (VWC)] in non-rainy days between 8:00-20:00 h during the mid-growing season of June 1-August 31 for 2013 and 2014. Data points are binned values from pooled data over two years at increments of 40 W $m^{-2}$, 1.2 °C, 0.3 kPa, and 0.005 $m^3\,m^{-3}$ for $R_s$, $T$, VPD and VWC, respectively.

**Fig. 4** Sap-flow rate per leaf area ($J_s$) in non-rainy, daytime hours during the mid-growing season of June 1-August 31 for both 2013 and 2014 as a function of shortwave radiation ($R_s$), air temperature ($T$), vapor pressure deficit (VPD) under high volumetric soil water content (VWC > 0.10 $m^3\,m^{-3}$ both in 2013 and 2014) and low VWC (< 0.10 $m^3\,m^{-3}$,2013 and 2014). $J_s$ is given as binned averages according to $R_s$, $T$, and VPD, based on increments of 100 W

m$^{-2}$, 1°C, and 0.2 kPa, respectively. Bars indicate standard error.
**Fig. 5** Regression slopes of linear fits between sap-flow rate per leaf area ($J_s$) in non-rainy
days and shortwave radiation ($R_s$), vapor pressure deficit (VPD), air temperature ($T$), and
volumetric soil water content (VWC) between 8:00-20:00 h during the mid-growing season
of June 1-August 31 for 2013 and 2014.
**Fig. 6** Mean monthly diurnal changes in sap-flow rate per leaf area ($J_s$) and stomatal
conductance ($g_s$) in *Artemisia ordosica* during the growing season (May-September) for both
2013 and 2014. Each point is given as the mean at specific times during each month.
**Fig. 7** Seasonal variation in hysteresis loops between sap-flow rate per leaf area ($J_s$) and
shortwave radiation ($R_s$) using normalized plots for both 2013 and 2014. The y-axis
represents the proportion of maximum $J_s$ (dimensionless), and the x-axis represents the
proportion of maximum $R_s$ (dimensionless). The curved arrows indicate the clockwise
direction of response during the day.
**Fig. 8** Sap-flow rate per leaf area ($J_s$) and shortwave radiation ($R_s$) over consecutive three
days in 2013, i.e., (a) under low volumetric soil water content (VWC) and high vapor pressure
deficit (VPD; DOY 153-155, VWC=0.064 m$^3$ m$^{-3}$, REW=0.025, VPD=2.11 kPa), (b)
moderate VWC and VPD (DOY 212-214, VWC=0.092 m$^3$ m$^{-3}$, REW=0.292, VPD=1.72
kPa), and (c) high VWC and low VPD (DOY 192-194, VWC=0.152 m$^3$ m$^{-3}$, REW=0.865,
VPD= 0.46 kPa). REW is the relative extractable soil water. VWC, REW, and VPD are the
mean value of the three days.
**Fig. 9** Time lag between sap-flow rate per leaf area ($J_s$) and short wave radiation ($R_s$) in
relation to volumetric soil water content (VWC). Hourly data in non-rainy days during the
mid-growing season of June 1-August 31 for 2013 and 2014. The lag hours were calculated
by a cross-correlation analysis using a three-day moving window with a one-day time step.
Rainy days were excluded. The solid line is based on exponential regression ($p<0.05$).
**Fig. 10** Relationship between volumetric soil water content (VWC) and (a) stomatal
conductance ($g_s$) in *Artemisia ordosica*, and (b) decoupling coefficient ($\Omega$) for 2013 and 2014.
Hourly values are given as binned averages based on a VWC-increment of 0.005 $m^3$ $m^{-3}$.
Bars indicate standard error. Only regressions with $p$-values $< 0.05$ are shown.


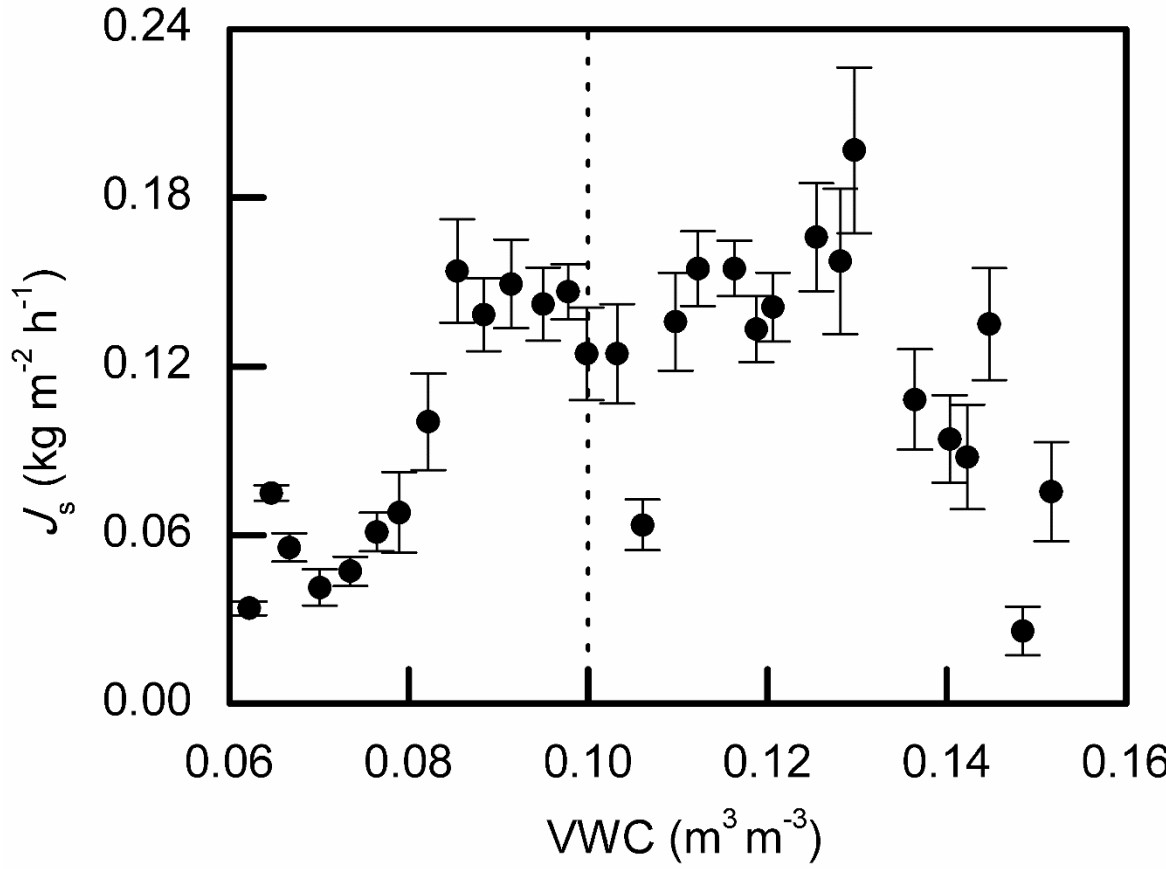


**Fig. 1** Sap-flow rate per leaf area ($J_s$) as a function of soil water content (VWC) at 30 cm
depth in non-rainy, daytime hours during the mid-growing period from June 1-August 31
over 2013-2014. Data points are binned values from pooled data over two years at a VWC
increment of 0.003 $m^3\,m^{-3}$. Dotted line represents the VWC threshold for $J_s$.

600 .



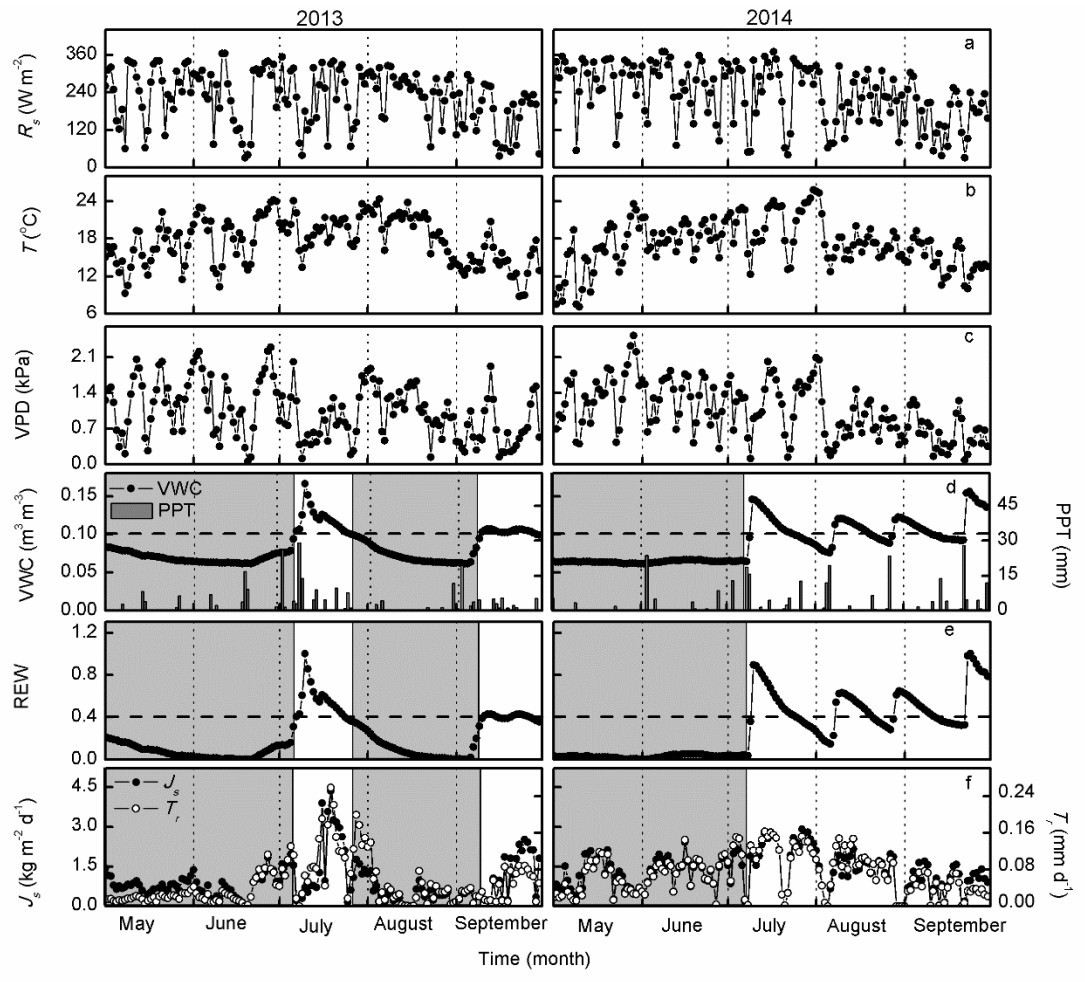



**Fig. 2** Seasonal changes in daily (24-hour) mean shortwave radiation ($R_s$; a), air temperature
($T$; b), vapor pressure deficit (VPD; c), volumetric soil water content (VWC; d), relative
extractable water (REW; e), daily total precipitation (PPT; d), and daily sap-flow per leaf
area ($J_s$; f), and daily transpiration ($T_r$, mm d$^{-1}$; f) from May to September for both 2013 and
2014. Horizontal dash lines (d, e) represent VWC and REW threshold of 0.1 m$^3$ m$^{-3}$ and 0.4,
respectively. Shaded bands indicate periods of drought.

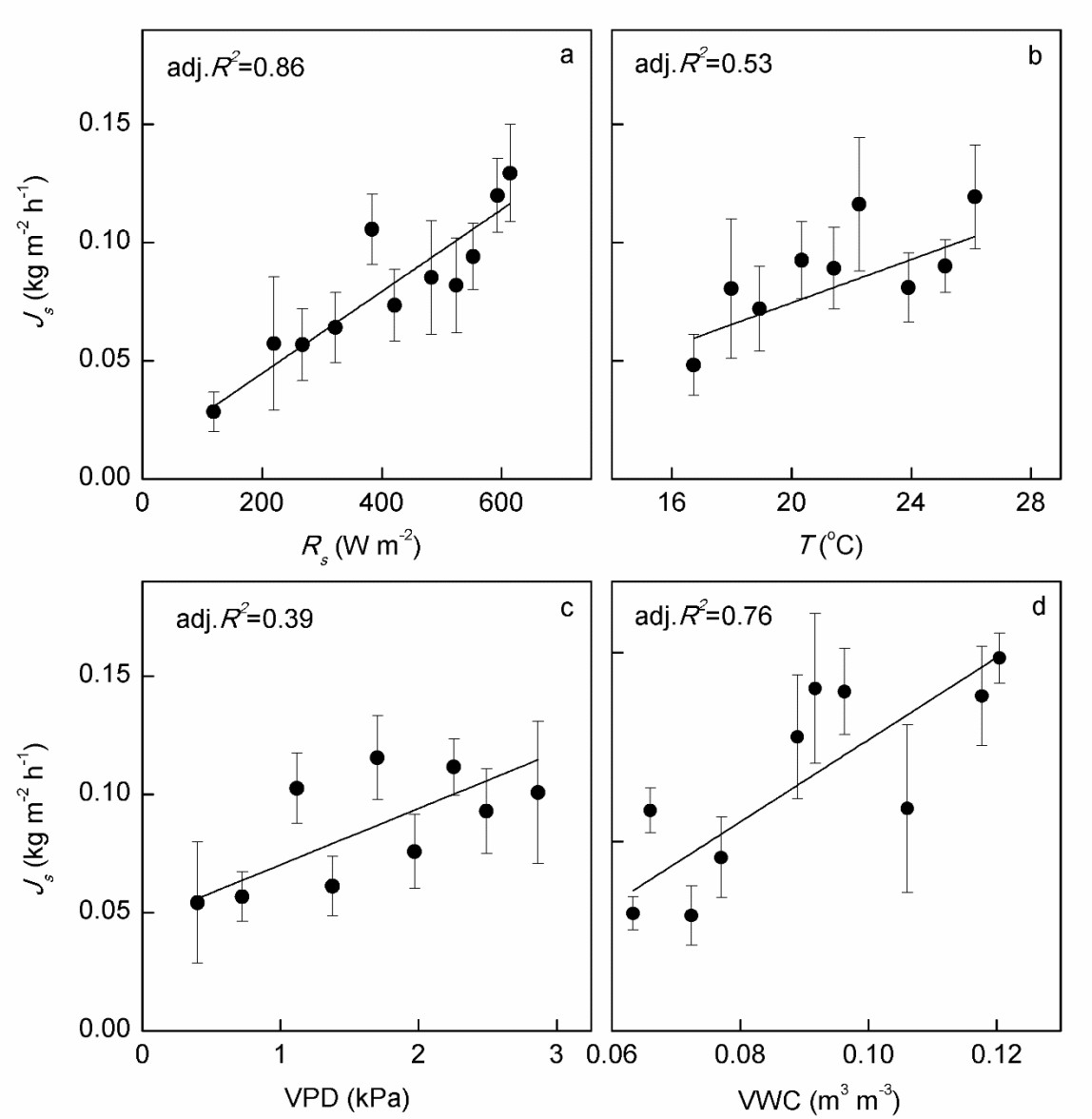



**Fig. 3** Relationships between sap-flow rate per leaf area ($J_s$) and environmental factors
[shortwave radiation ($R_s$), air temperature ($T$), vapor pressure deficit (VPD), and soil water
content at 30-cm depth (VWC)] in non-rainy days between 8:00-20:00 h during the mid-
growing season of June 1-August 31 for 2013 and 2014. Data points are binned values from
pooled data over two years at increments of 40 W m$^{-2}$, 1.2 °C, 0.3 kPa, and 0.005 m$^3$ m$^{-3}$ for
$R_s$, $T$, VPD and VWC, respectively.


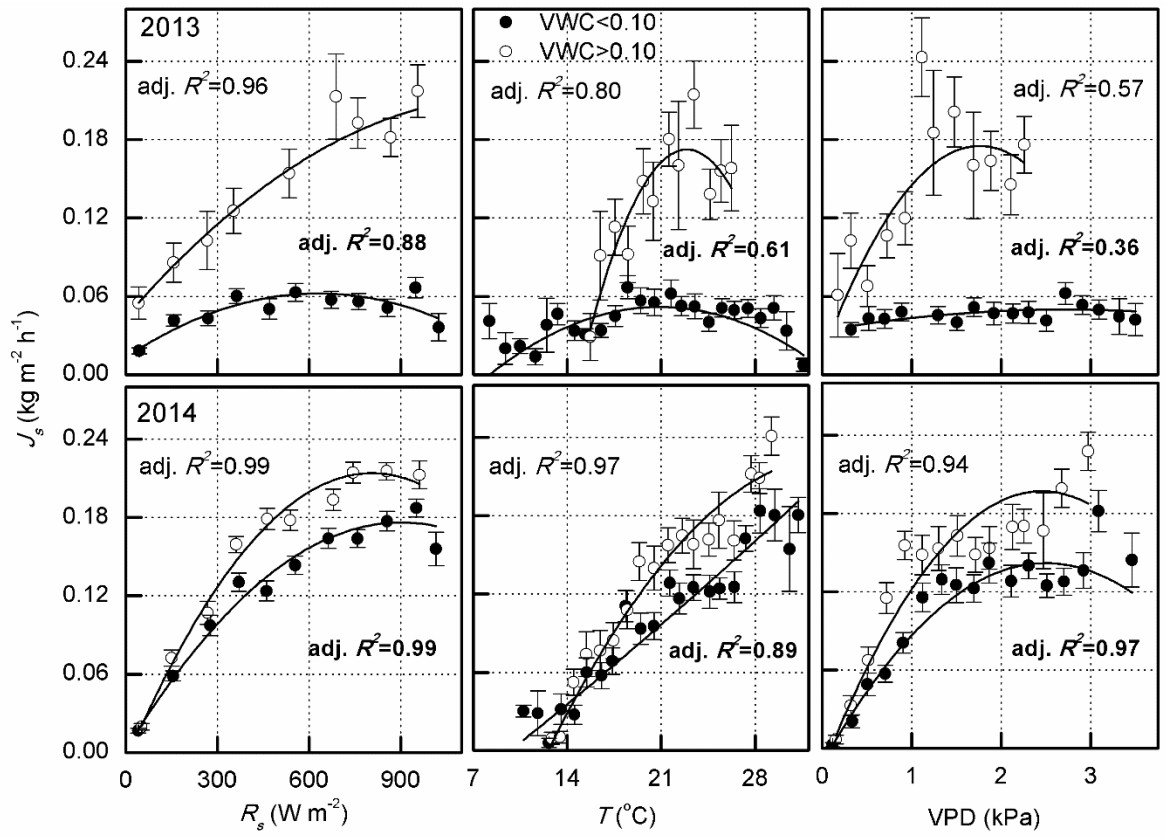



**Fig. 4** Sap-flow rate per leaf area ($J_s$) in non-rainy, daytime hours during the mid-growing


season of June 1-August 31 for both 2013 and 2014 as a function of shortwave radiation ($R_s$),


air temperature ($T$), vapor pressure deficit (VPD) under high volumetric soil water content


(VWC > 0.10 $m^3$ $m^{-3}$ both in 2013 and 2014) and low VWC (< 0.10 $m^3$ $m^{-3}$, 2013 and 2014).


$J_s$ is given as binned averages according to $R_s$, $T$, and VPD, based on increments of 100 W


$m^{-2}$, 1°C, and 0.2 kPa, respectively. Bars indicate standard error.




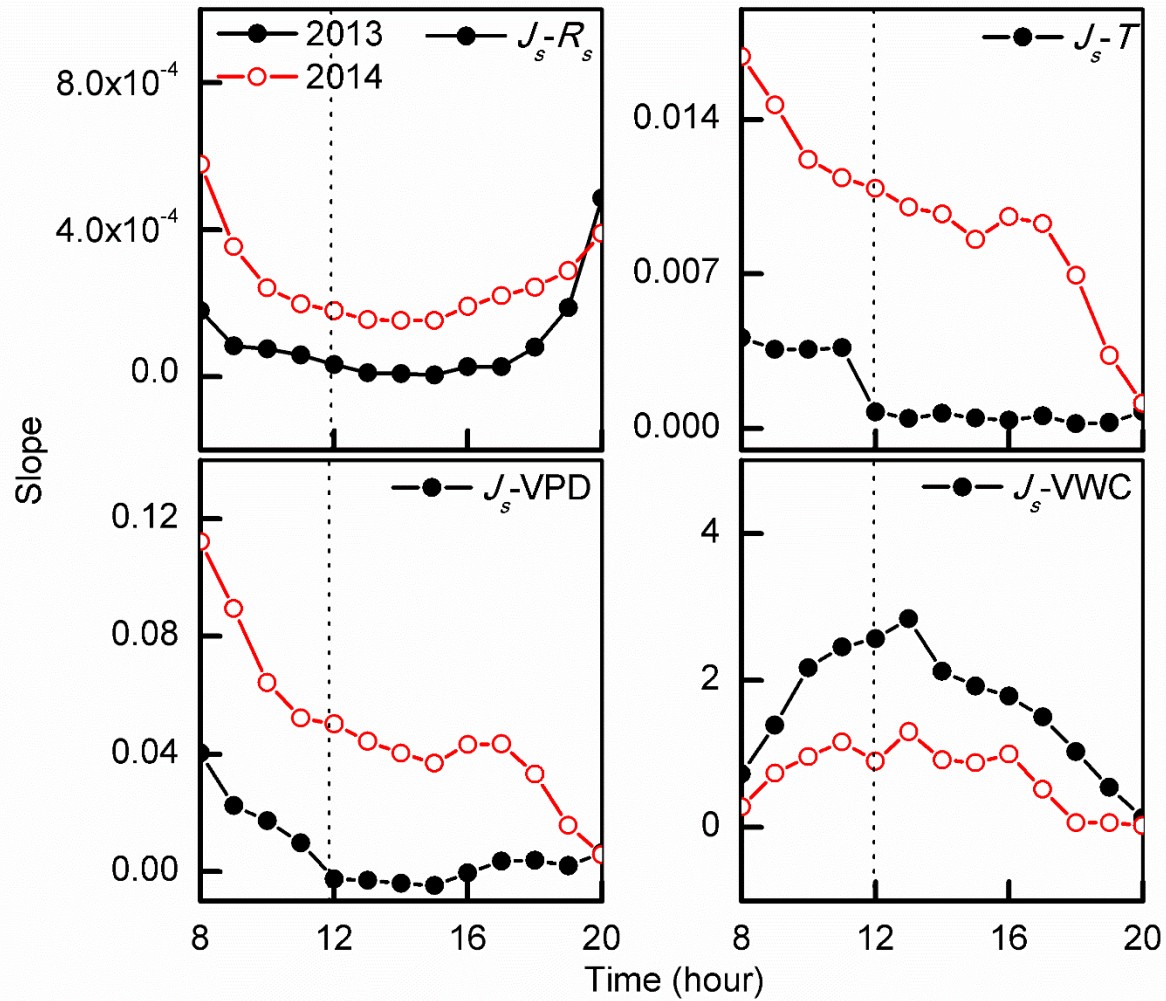


**Fig. 5** Regression slopes of linear fits between sap-flow rate per leaf area ($J_s$) in non-rainy

days and shortwave radiation ($R_s$), vapor pressure deficit (VPD), air temperature ($T$), and

volumetric soil water content (VWC) between 8:00-20:00 h during the mid-growing season

of June 1-August 31 for 2013 and 2014.


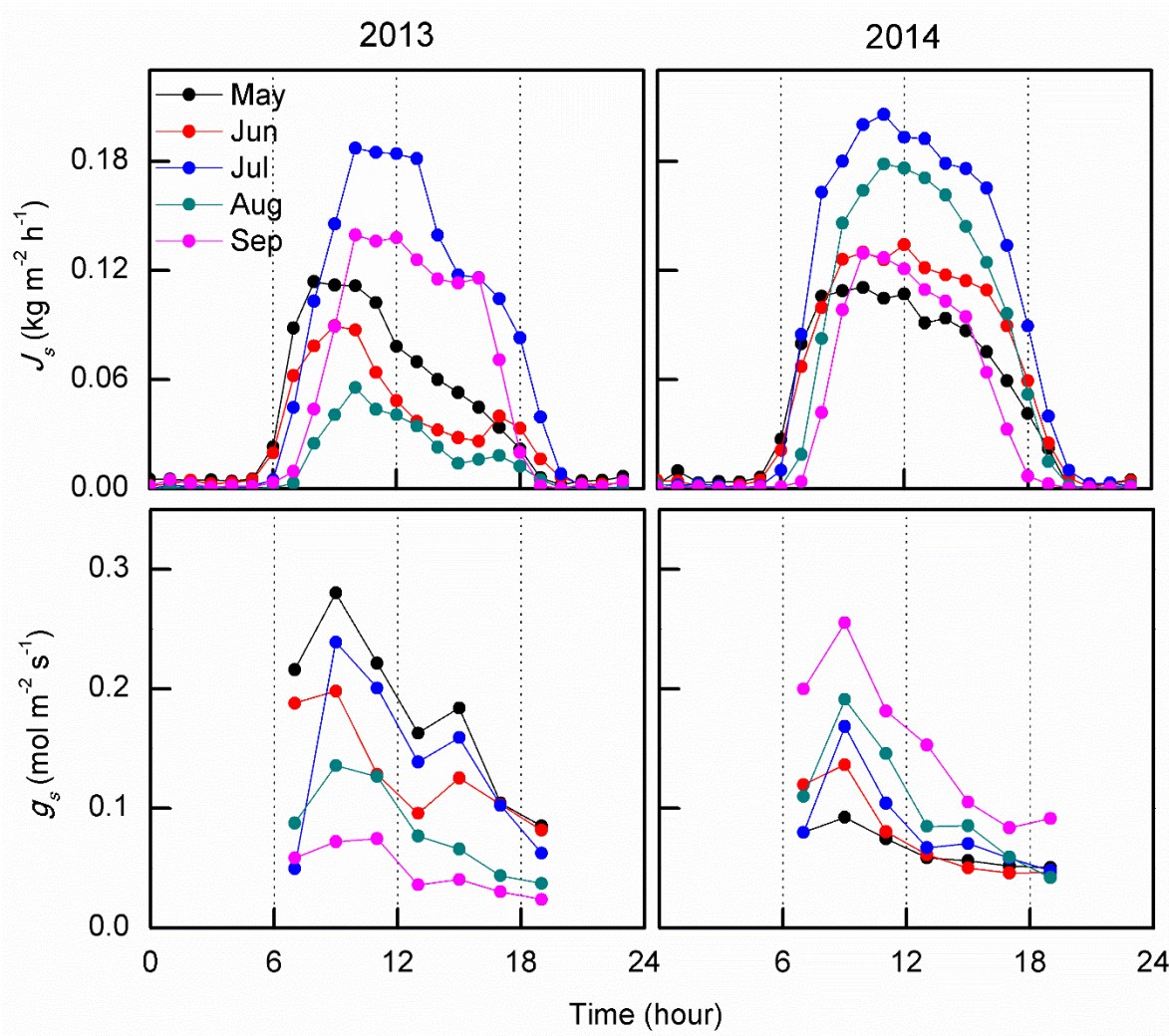



**Fig. 6** Mean monthly diurnal changes in sap-flow rate per leaf area ($J_s$) and stomatal conductance ($g_s$) in *Artemisia ordosica* during the growing season (May-September) for both 2013 and 2014. Each point is given as the mean at specific times during each month.




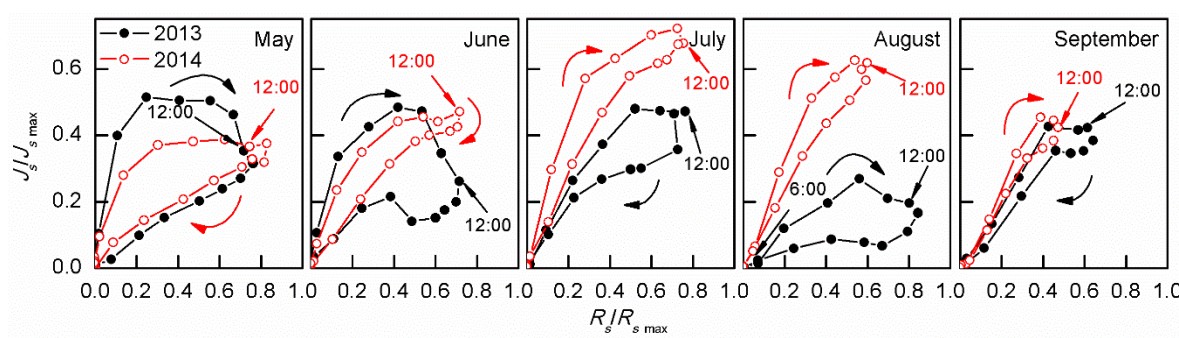



**Fig. 7** Seasonal variation in hysteresis loops between sap-flow rate per leaf area ($J_s$) and
shortwave radiation ($R_s$) using normalized plots for both 2013 and 2014. The y-axis
represents the proportion of maximum $J_s$ (dimensionless), and the x-axis represents the
proportion of maximum $R_s$ (dimensionless). The curved arrows indicate the clockwise
direction of response during the day.




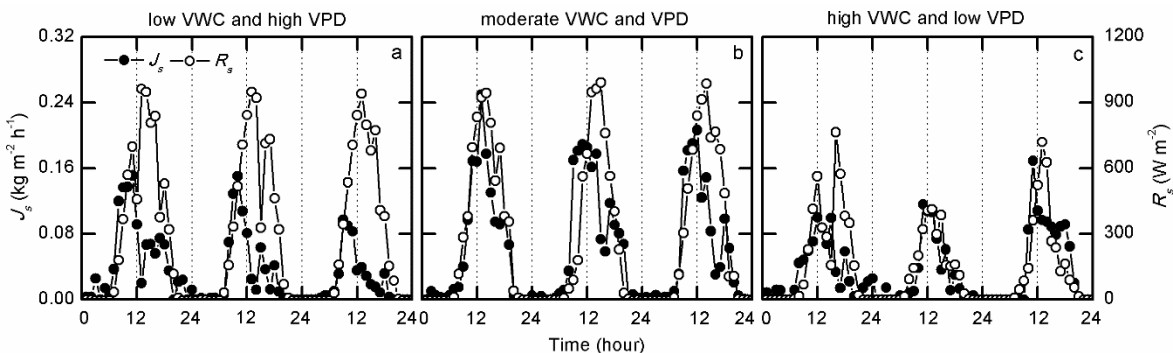



**Fig. 8** Sap-flow rate per leaf area ($J_s$) and shortwave radiation ($R_s$) over consecutive three
days in 2013, i.e., (a) under low volumetric soil water content (VWC) and high vapor pressure
deficit (VPD; DOY 153-155, VWC=0.064 $m^3$ $m^{-3}$, REW=0.025, VPD=2.11 kPa), (b)
moderate VWC and VPD (DOY 212-214, VWC=0.092 $m^3$ $m^{-3}$, REW=0.292, VPD=1.72
kPa), and (c) high VWC and low VPD (DOY 192-194, VWC=0.152 $m^3$ $m^{-3}$, REW=0.865,
VPD= 0.46 kPa). REW is the relative extractable soil water. VWC, REW, and VPD are the
mean value of the three days.

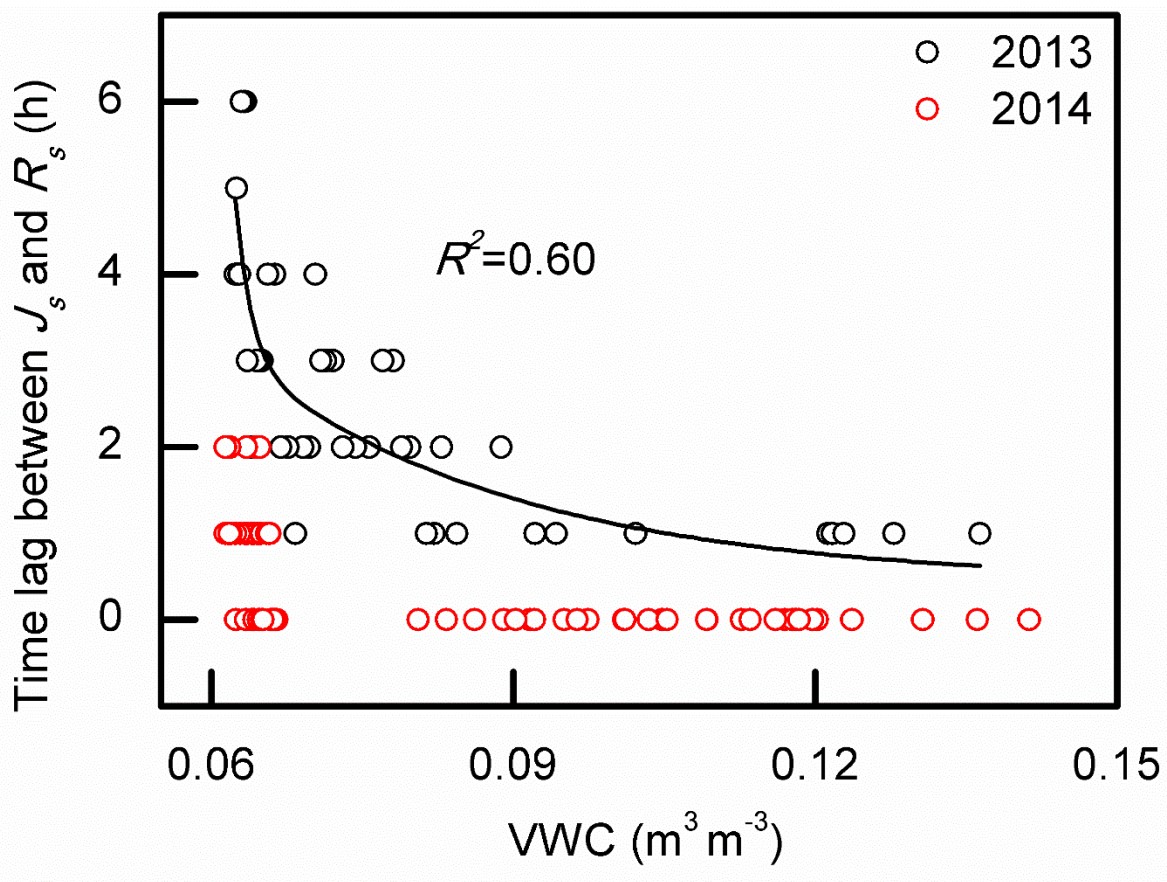



**Fig. 9** Time lag between sap-flow rate per leaf area ($J_s$) and short wave radiation ($R_s$) in
relation to volumetric soil water content (VWC). Hourly data in non-rainy days during the
mid-growing season of June 1-August 31 for 2013 and 2014. The lag hours were calculated
by a cross-correlation analysis using a three-day moving window with a one-day time step.
Rainy days were excluded. The solid line is based on exponential regression ($p<0.05$).


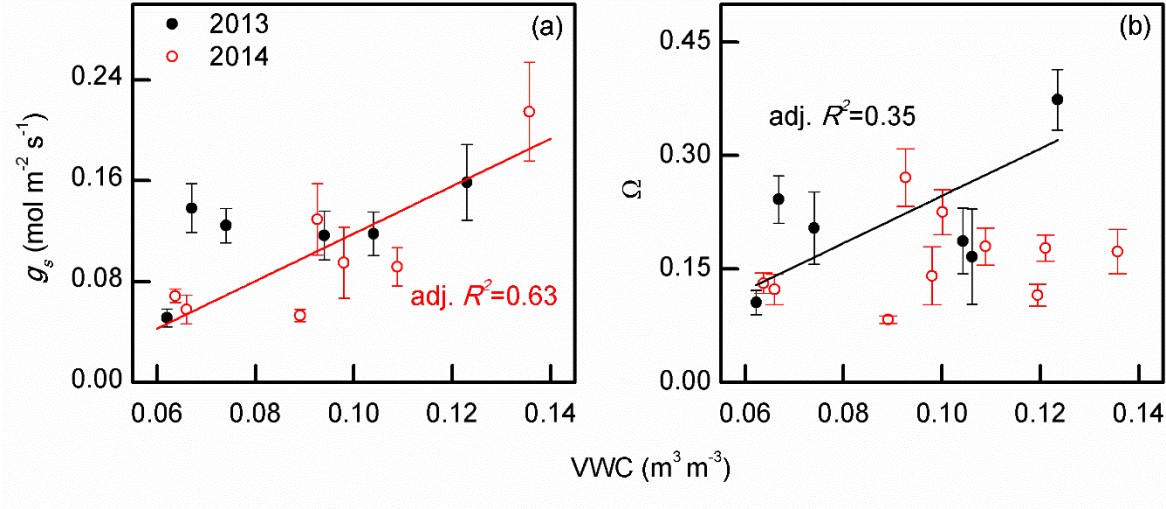



**Fig. 10** Relationship between volumetric soil water content (VWC) and (a) stomatal conductance ($g_s$) in *Artemisia ordosica*, and (b) decoupling coefficient ($\Omega$) for 2013 and 2014. Hourly values are given as binned averages based on a VWC-increment of $0.005$ m$^3$ m$^{-3}$. Bars indicate standard error. Only regressions with $p$-values $< 0.05$ are shown.

683