# Peer review of "Soil moisture control on sap-flow response to biophysical factors in a desert-shrub species, *Artemisia ordosica"

_Biogeosciences, 2016_

## Referee Comment (RC1) · Anonymous Referee #1 · 13 Dec 2016

General comments

The paper presents an interesting analysis of soil moisture control on the response of sap-flow to biophysical factors in a desert-shrub species. As sap-flow is a powerful indicator of water transport in the soil-plant-atmosphere system, understanding the effect of dry conditions on sap flow is very valuable. The topic is suitable for the journal. However the analysis can go deeper as suggested below. The clarity of the analysis and writing should be improved.

*As the paper is about the soil moisture control on sap-flow and its response to meteorological variables, a physical basis for the definition for drought condition and its severity should be included. Instead, the authors keep on changing their definition of

dry conditions for each year and in various figures. What is the reason for using 0.08 m3 m-3, as the threshold to identify drought periods? Is it for severe, moderate or mild drought? The analysis lacks consistency (Example figures 2 6). In section 2.4, they used 0.08 m3 m-3 as the threshold to identify drought conditions. In Figure 2, it is 0.11 m3 m-3 for the drought year 2013 and 0.09 for the wet 2014. Why don't they use 0.08 m3 m-3 in both years? They change threshold, definition of dry condition and VWC values in figure 6. I strongly suggest being consistent in their definition of drought conditions and use the same threshold in all figures.

*The root zone depth for this species is around 60cm (Line 291). The water deep in the root zone can maintain transpiration rates even at low VWC. I think a better way is to define threshold based on root zone soil water content in this paper. Is there any field capacity or wilting point measurements available at the site? If so mention that in the paper and use relative available water content in the root zone. If not, use relative water content (based on maximum and minimum VWC values at the site) in the 30 cm soil layer to identify the drought conditions. The value of VWC shown in Figure 1 indicate that soil drying occurred mainly in shallow layer, not in the deep layer (30 cm), especially during pre and post growing periods.

*Is it possible to include transpiration (mm) values in this paper? That will add more value to understand the acclimation process of plants to the dry conditions.

*The methods section reports leaf area measurements, but its values are not mentioned in the paper. Even in conclusion they have mentioned leaf expanded periods, but no data to support it. *Statistical significance should be evaluated for each figure and include p value along with R2 in figures. Even though authors say they used p=0.05 (Line 185), the value of R2 and number of points suggests some relations are not statistically significant. All those figures that shows statistically significant results (p<0.05) only should be included in the paper, else should be removed (see specific comments below).

*The 2013-2014 data shown clearly indicate that there is no direct control of VWC on sap flow (Figure 2 first vertical panel). Figure 2 also clearly shows that the relation between Js with Rs , T and VPD are non-linear (check previous comments on p-value). If the relationships are non-linear how can they explain the linear regression slopes shown in Figure 3? Is the linear relationships shown are statistically significant?

Specific Comments

Abstract: 0.11 m3 m-3 is only for 2013, not for 2014.

Introduction: The section need to highlight what is the need for sap-flow measurements and how it influence ecosystem water transport and balance. The importance and need for the study is not properly addressed even though the authors explain the effect of environmental variables on sap-flow in this section. In addition to this the section should refer more recent papers on sap-flow measurements.

Line 131: Also include root zone depth, and mean leaf area values. Is it possible to include field capacity and wilting point here?

Line 140: 'after dynamax 2005', what is that?

Line 141: What is the frequency of measurements?

Line 143: What was the mean leaf area? How did it vary with season?

Line 151-155: decoupling coefficient re-expresses gs, and can be removed.

Line 164-171: Be consistent with label style. It is better to italicize all mathematical variable labels. Only u, gs and Js are italicized

Line 178- What is the reason for selecting VWC =0.08 m3 m-3 as the threshold to determine the drought condition. It is not explained in the paper. The time series of VWC (Figure 1) don't show any severe drought conditions in 30 cm depth. It is useful if the authors can include relative water content within 0-30 cm layer.

[Figure]

Line 183: Is it linear regression? See general comments on the same.

Line 197: 'Lower than. . .' What is the reduction in percentage?

Line 201-204: See general comments above.

Line 205-210: Can you add time series of gs here?

Line 215: See general comments above. The threshold to define drought conditions should be the same in both years

Line 222-228: How can you explain the use of the slope of linear regression relationship if the variation of Js with Rs, T and D are non-linear? Use values only when p<0.05

Line 233-243: What is the reason for this delay?

Line 246-252: Figure 6 shows data from three days. What are those days in day number? A meaning full explanation should be given for the use of VWC limits. The first panel shows VWC variation within 0.001 m3 m-3! Is it meaning full considering the errors in VWC measurements? Also use only significant digits while using VWC values. The data is only three days. Is it possible to add more data in this figure, also from both years? Using only three days for this analysis is not conclusive.

Line 252: Figure 8 should be included in the results section.

Line 263-265: This is already known. Provide some references here.

Line 271: Rewrite this sentence. VWC don't show any direct effect on Js in the figure shown

Line 288:'rate of transpiration' See general comments on the inclusion of transpiration

Line 291: Provide information on root zone depth in the methods section.

Figure 1: Dotted line is not explained in the figure caption.

Figure 2: Use the same definition for dry periods in 2013 and 2014 as mentioned

above.

Figure 6: Use only most significant digits for VWC. See the comments above on the consistency on the definition of dry conditions. The data is shown for only three days. Is it possible to include more data like a month or more from both the years?

Figure 8: This figure is not mentioned in results sections. Look like p value is low (both N and R2 low) and not statistically significant. If it is below 95

Conclusion: It should be rewritten based on the comments above.

---

## Referee Comment (RC2) · Anonymous Referee #2 · 24 Dec 2016

The study investigates the modulation of sap-flow response to biophysical factors (solar radiation, air temperature, and vapor pressure deficit) through soil moisture content in a semi-arid ecosystem. Soil moisture controls sap flux in shrub, Artemisia ordosica. Diurnal course of sap flow to biophysical factors modulated by soil moisture contents, and sensitivity enhanced with higher soil moisture levels.

I enjoyed reading. It will be a good contribution to Biogeosciences. I have some minor comments which should be addressed about the drought classification, soil and vegetation characteristics. Also at the end, I have some minor points about writing.

Based on Li et al., 2014 (L108), shrub is shallow rooted. If available/known, it may be good to include root distribution of Artemisia ordosica such as XX% of shrub roots are

located within the top 30 cm, and tap root can reach up to 60 cm (Zhao et al., 2010 from L291). This also supports your soil moisture content measurements in the top 10 cm and 30 cm.

If available, it will be good to include stomata closure point, wilting point, and hygroscopic point levels. Hence, the reader can judge the severity of drought. So, there will be some justification based on your drought classification. You used 0.08 (L178), 0.09 (Figure 2), and 0.11 (Figure 2). Is it 0.08 or 0.09?

Also knowing wilting point and hygroscopic point helps us appreciating the Figure 6. You stratified soil water content based on three limits. How much severe the lowest value. My back of envelope calculation by using Campbell (1974) for sandy soil where porosity is $\sim$0.42 (1-1.54/2.65), the wilting point (15000cm) is $\sim$0.07. It seems your wilting point is much lower. Definitely, to appreciate the Figure 6 and drought severity, giving values are beneficial.

A little more detail about vegetation setting is beneficial. LAI and plant canopy cover of shrub are beneficial. As far as I know, in Mu Us Desert dunes are migrating or semi-migrating depending on canopy cover. So, it will be beneficial for readers.

L272-274. In your DISCUSSION, it will be good to include climate for these plant species. Because your ecosystem which is water-limited, most probably different than their study sites! For example, Huang et al. (2009) study site (L275) is in Guangxi, China where annual precipitation is 1900 mm, and mean annual temperature is 19.3°C. Most probably some/most part of the year, the ecosystem is energy-limited. So, it is not so surprising to see solar radiation control on sap flow. I could not find electronic copy of Zhang et al. (2003) work. Please include prevailing climate in their study area too.

L276-L278. To emphasize the importance of small events on ecological processes, I want to draw authors attention another study by Sala and Lauenroth (1982). Sala and Lauenroth (1982) showed the ecological importance of small events (<5mm) in

semiarid site where dominated by C4 grass. I will be worth to check!

Sala O.E. and W.K. Lauenroth (1982). Small rainfall events: an ecological role in semi-arid regions. Oecologia, 53 (3), 301-304.

Minor Points:

L69. VERB. . . ..low soil water availability limitS . . ...,

L70. VERB. . . ...limitS vegetation productivity

L73. I recommend citation for: grass replacement by shrubs.

L103. Capitalization. . . .the Mu Us Desert. . ...

L125. Capitalization. . . .the Mu Us Desert. . ...

L137. VERB. Mean height and sapwood area of sampled shrubs WERE . . ...

L156. Replace UPSILON in the equation with lower-case gamma, $\gamma$ for psychrometric constant.

L161. Insert a comma after "ground". . . ..the ground, and. . ..

L180. VERB. Linear and nonlinear regression WERE . . ...

L197. VERB. Total precipitation and number of rainfall events. . .. WERE lower than THOSE. . ..

L266. VERB. Synergistic interactions . . ... ARE. . ..

L355-461. Please go through the references. Make sure the unity within the references. Journal names abbreviated some of them (L361, L367, L370 etc.), but not others (L 358, L383, L386 etc.). Choose one of them and stick with it. L424. Typo. Systems. . . L430. Typo. EcologY. . . L449. Typo. PLoS ONE. Compare with (L461 and L373). Use lower case for article names. Check (L456, L461, L416 etc.).

L541. Figure 4. I recommend following some color scheme (pattern) to represent

different months such as jet etc. This change will help the readers to follow the figure easier than the current form.

L553. Insert a comma after (dimensionless). . . .. (dimensionless), and . . ..

L554. To distinguish from straight arrows, I recommend using 'curved arrows' such as: The CURVED arrowS indicate the clockwise. . ..

L558. I recommend using 'three' instead of '3' days.

---

## Author Comment (AC1) · 3 Feb 2017

**Response list to the reviewers' comments**

Ref: doi:10.5194/bg-2016-480, 2016

Title: **Soil moisture control on sap-flow response to biophysical factors in a desert-shrub species, Artemisia ordosica**

Authors: TianShan Zha, Duo Qian, Xin Jia, Yujie Bai, Yun Tian , Charles P.-A. Bourque, Jingyong Ma, Wei Feng, Bin Wu, Heli Peltola

**Dear Editor,**

**Thank you very much for your helpful comments and suggestions for improvement of our manuscript. We have been carefully looking at your comments and revising the manuscript accordingly. The following below are our responses showing how we have been revising the manuscript.**

**We are looking forward to your further comments and a possible publication in the BG special issue** (Ecosystem processes and functioning across current and future dryness gradients in arid and semi-arid lands)**.**

**Kind regards,**

**Tianshan Zha**
* * *
**Anonymous Referee #1**

General comments

*As the paper is about the soil moisture control on sap-flow and its response to meteorological variables, a physical basis for the definition for drought condition and its severity should be included. Instead, the authors keep on changing their definition of dry conditions for each year and in various figures. What is the reason for using 0.08 m3 m-3, as the threshold to identify drought periods? Is it for severe, moderate or mild drought? The analysis lacks consistency (Example figures 2 6). In section 2.4, they used 0.08 m3 m-3 as the threshold to identify drought conditions. In Figure 2, it is 0.11 $m^3$ $m^{-3}$ for the drought year 2013 and 0.09 for the wet 2014. Why don't they use 0.08 $m^3$ $m^{-3}$ in both years? They change threshold, definition of dry condition and VWC values in figure 6. I strongly suggest being consistent in their definition of drought conditions and use the same threshold in all figures. *The root zone depth for this species is around 60 cm (Line 291). The water deep in the root zone can maintain transpiration rates even at low VWC. I think a better way is to define threshold based on root zone soil water content in this paper. Is there any field capacity or wilting point measurements available at the site? If so mention that in the paper and use relative available water content in the root zone. If not, use relative water content (based on maximum and minimum VWC values at the site) in the 30 cm soil layer to identify the drought conditions. The value of VWC shown in Figure 1 indicate that soil drying occurred mainly in shallow layer, not in the deep layer (30 cm), especially during pre and post growing periods.

RE: We agree with reviewer's suggestion of a consistent threshold of soil water content among years. We will be defining the soil drought conditions based on relative extractable soil water (REW) at 30cm depth during measuring period (2013-2014) in the revised manuscript. The consistent soil water threshold for sapflow will be taken over years in the revised manuscript. From our preliminary analysis, the plant is in drought condition when VWC at 30 cm depth is 0.10 $m^3$ $m^{-3}$ which is equivalent to a REW value of 0.4 that was proposed by Granier et al. (1999;

2003).

Literatures that will be added in the revised manuscript:

1. Granier et al.: Evidence for soil water control on carbon and water dynamics in European forests during the extremely dry year: 2003. Agricultural and Forest Meteorology, 143,123-145, 2007.

2. Granier, A., Bre´da, N., Biron, P., Villette, S.: A lumped water balance model to evaluate duration and intensity of drought constraints in forest stands. Ecol. Model. 116, 269–283, 1999.

*Is it possible to include transpiration (mm) values in this paper? That will add more value to understand the acclimation process of plants to the dry conditions.

RE: Yes, we will add estimated transpiration (mm) values in the revised manuscript. The transpiration has been estimated on the basis of leaf area index (LAI) and sapflow per leaf area.

*The methods section reports leaf area measurements, but its values are not mentioned in the paper. Even in conclusion they have mentioned leaf expanded periods, but no data to support it.

RE: The leaf area measurements were used to calculate the sap flow rate per leaf area, which is a comparative unit with other species. We will add the equation for calculating sap flow rate per leaf area in the revised manuscript. Leaf area measurements will be also added to revised manuscript.

We will also add observations of phenological phases in revised manuscript. Phenological phases observation has been briefly described in "2.1 Experimental site", Line 129-130 "Normally, shrub leaf-expansion, leaf-expanded, and leaf-coloration stages begin in April, June, and September, respectively."

*Statistical significance should be evaluated for each figure and include p value along with R2 in figures.

RE: We will add statistical significance and R square value to figures in the revised manuscript.

*The 2013-2014 data shown clearly indicate that there is no direct control of VWC on sap flow (Figure 2 first vertical panel). Figure 2 also clearly shows that the relation between Js with Rs , T and VPD are non-linear (check previous comments on p-value). If the relationships are non-linear how can they explain the linear regression slopes shown in Figure 3? Is the linear relationships shown are statistically significant?

RE: We defined a VWC cut-point (Figure 2 first vertical panel) as a starting VWC value of drought.  We will redefine VWC threshold of sapflow in revised manuscript using REW value. From our preliminary analysis, the VWC threshold of 0.10 $m^3m^{-3}$ is equivalent to a REW threshold of 0.4 (Granier et al., 1999, 2002; Bernier et al. 2002).

Each slope value in Fig. 3 were from a regression between sap-flow rates at specific time and its corresponding environmental factor over growing season (Jun. 1 to Aug. 31). This regression is more like linear relationship and statistically significant. We will show this regression and p-values in revised manuscript.

References:

1. Granier et al.: Evidence for soil water control on carbon and water dynamics in European forests during the extremely dry year: 2003. Agricultural and Forest Meteorology, 143,123-145, 2007.

2. Granier, A., Bre´da, N., Biron, P., Villette, S.: A lumped water balance model to evaluate duration and intensity of drought constraints in forest stands. Ecol. Model. 116, 269–283, 1999.

3. Bernier, P.Y., Bre´da, N., Granier, A., Raulier, F., Mathieu, F.: Validation of a canopy gas exchange model and derivation of a soil water modifier for transpiration for sugar maple (Acer saccharum Marsh.) using sap flow density measurements. For. Ecol. Manage. 163, 185–196, 2002.

Specific Comments

*Abstract: 0.11 m3 m-3 is only for 2013, not for 2014.

RE:We will use consistent VWC threshold of 0.10 for both years as specified above. This will be revised in the revised manuscript.

*Introduction: The section need to highlight what is the need for sap-flow measurements and how it influence ecosystem water transport and balance. The importance and need for the study is not properly addressed even though the authors explain the effect of environmental variables on sap-flow in this section. In addition to this the section should refer more recent papers on sap-flow measurements.

RE: We will be adding the statements addressing the importance and need for the sap flow study in revised manuscript.

*Line 131: Also include root zone depth, and mean leaf area values. Is it possible to include field capacity and wilting point here?

RE:The root zone depth and leaf area values will be added into revised manuscript.

There was no data for field capacity and wilting point values available. Therefore, we calculated the relative extractable soil water (REW) as suggested by reviewer and literature (Fernández et al., 1997; Zeppel et al., 2008).

$REW=(VWC-VWC_{min})/(VWC_{max}-VWC_{min})$,

where VWC is daily soil water content ($m^3 m^{-3}$), $VWC_{min}$ and $VWC_{max}$ are the minimum and maximum VWC during the measurement period in each year, respectively.

References:

1. Fernández, J. E., Moreno, F., Girón I. F., and Blázquez, O. M.: Stomatal control of water use in olive tree leaves, Plant Soil, 190, 179–192, 1997.

2. Zeppel, M. J. B., Macinnis-Ng, C. M. O., Yunusa, I. A. M., Whitley, R. J., Eamus, D. Long term trends of stand transpiration in a remnant forest during wet and dry years, J. Hydrol., 349, 200-213, 2008.

*Line 140: 'after dynamax 2005', what is that?

RE: This citation 'after dynamax 2005' will be revised into 'Dynamax, 2005'.

*Line 141: What is the frequency of measurements?

RE:The frequency of sapflow measurements is one record per minute. This will be added to the method section in revised manuscript.

*Line 143: What was the mean leaf area? How did it vary with season?

RE: Mean leaf area is mean of estimated leaf areas of five shrubs. The leaf area of each shrub is

the product of branch numbers and leaf area per branch. The seasonal changes in leaf area will be present in revised manuscript.

*Line 151-155: decoupling coefficient re-expresses gs, and can be removed.
RE:We will be removing decoupling coefficient in revised manuscript.

*Line 164-171: Be consistent with label style. It is better to italicize all mathematical variable labels. Only u, gs and Js are italicized.
RE:We will consistently italicize the term label throughout the text in revised manuscript.

*Line 178- What is the reason for selecting VWC=0.08 m3 m-3 as the threshold to determine the drought condition. It is not explained in the paper. The time series of VWC (Figure 1) don't show any severe drought conditions in 30 cm depth. It is useful if the authors can include relative water content within 0-30 cm layer.
RE: The VWC threshold of 0.08 was selected from first vertical panels of figure 2. The VWC threshold indicated the starting point of dry condition (or drought start). We agree with reviewer that this definition of threshold was not robust. We will revise the VWC threshold using relative extractable soil water (REW) and VWC at 30cm during measuring period (2013-2014) in revised manuscript.

*Line 197: 'Lower than...' What is the reduction in percentage?
RE:We will add the statement "Total precipitation and number of rainfall events during the 2013 measurement period (257.2 mm and 46 days) were about 5.6% and 9.8% lower than those during 2014 (272.4 mm and 51 days; Fig, 1d), respectively." into revised manuscript.

*Line 205-210: Can you add time series of gs here?
RE:Yes, we will add time series of gs as suggested by reviewer.

*Line 215: See general comments above. The threshold to define drought conditions should be the same in both years.
RE:Yes, we will revise VWC threshold as responded to general comments above.

*Line 222-228: How can you explain the use of the slope of linear regression relationship if the variation of Js with Rs, T and D are non-linear? Use values only when p<0.05.
RE: The each of slope values in Fig. 3 was from a regression between sap-flow rates at specific time and its corresponding environmental factor over growing season (Jun. 1 to Aug. 31). This regression is close to linear relationship and statistically significant. We will show this regression and p-values in revised manuscript.

*Line 233-243: What is the reason for this delay?
RE:The reason for the delay between Js and $R_s$ would be energy force. Specific explanation would be that $R_s$ can force T and VPD to increase, causing a phase difference in time lags among the relations $J_s$-$R_s$, $J_s$-T, and $J_s$-VPD. Stomatal conductance gs peak earlier than Rs. These delay reflect an acclimation of plant to dry and hot environment. We will add detailed explanation in the

revised manuscript.

*Line 246-252: Figure 6 shows data from three days. What are those days in day number? A meaning full explanation should be given for the use of VWC limits. The first panel shows VWC variation within 0.001 m3 m-3! Is it meaning full considering the errors in VWC measurements? Also use only significant digits while using VWC values. The data is only three days. Is it possible to add more data in this figure, also from both years? Using only three days for this analysis is not conclusive.

RE:We will add the DOY in Fig.6 caption. This figure compared the degree of soil water control on the hysteresis between $J_s$ and $R_s$. Therefore, we took three contrastive drought degrees (severe, moderate, light) to see corresponding changes in the hysteresis in response to the drought degrees. We will specify these and give more days for this analysis to make this result more conclusive in the revised manuscript.

*Line 252: Figure 8 should be included in the results section.
RE:We will include Fig.8 in the results section.

*Line 263-265: This is already known. Provide some references here.
RE: We will add references listed below to explain this result. Similar result was reported previously (Qian et al., 2015; Zha et al., 2013).
References that will be added:
1. Qian, D., Zha, T., Jia, X., Wu, B., Zhang, Y., Bourque C. P. A., Qin, S., and Peltola, H.: Adaptive, water-conserving strategies in Hedysarum mongolicum endemic to a desert shrubland ecosystem, Environ. Earth. Sci., 74, 6039–6046, 2015.
2. Zha, T., Li, C., Kellomäki, S., Peltola, H., Wang, K.-Y., and Zhang, Y.: Controls of Evapotranspiration and CO2 Fluxes from Scots Pine by Surface Conductance and Abiotic Factors, Plos One, 8, e69027, 2013.

*Line 271: Rewrite this sentence. VWC don't show any direct effect on Js in the figure shown
RE:We will revise this sentence as "VWC is the most important factor modifying the response of sap flow to other environmental factors."

*Line 291: Provide information on root zone depth in the methods section.
RE:We will add information on root zone depth in the methods section of revised manuscript. Over 60% of the total roots distributed in the 0-60cm depth (Zhao et al. 2010). Our measurements showed over 80% roots distributed in the 0-60cm depth (Jia et al., 2016).
References that will be added:
1. Zhao, W., Liu, B., Chang, X., Yang, Q., Yang, Y., Liu Z., Cleverly, J., Eamus, Derek.: Evapotranspiration partitioning, stomatal conductance, and components of the water balance: A special case of a desert ecosystem in China. J. Hydrol., 538, 374-386, 2016.
2. Xin Jia, Tianshan Zha, Jinnan Gong, Ben Wang, et al.: Carbon and water exchange over a temperate semi-arid shrublandduring three years of contrasting precipitation and soil moisturepatterns. Agricultural and Forest Meteorology, 228, 120-129, 2016.

*Figure 1: Dotted line is not explained in the figure caption.

RE:We will add the explanation of dotted line in the Fig.1 caption as " Dotted line represents VWC threshold of 0.10 $m^3$ $m^{-3}$ in the revised manuscript".

*Figure 2: Use the same definition for dry periods in 2013 and 2014 as mentioned above.

RE:We will revise VWC threshold and use the same definition for dry periods in 2013 and 2014 in the revised manuscript.

*Figure 8: This figure is not mentioned in results sections. Look like p value is low (both N and R2 low) and not statistically significant. If it is below 95.

RE: Fig.8 was simply used to explain the hysteresis between sapflow and environmental factors. We will add some results from figure 8 in the result section in the revised manuscript. The statistical significance has been checked. Only regression line with p value < 0.05 will be showed in this figure in the revised manuscript.
* * *
**Anonymous Referee #2**

*General comments

I have some minor comments which should be addressed about the drought classification, soil and vegetation characteristics. Also at the end, I have some minor points about writing. Based on Li et al., 2014 (L108), shrub is shallow rooted. If available/known, it may be good to include root distribution of Artemisia ordosica such as XX% of shrub roots are located within the top 30 cm, and tap root can reach up to 60 cm (Zhao et al., 2010 from L291). This also supports your soil moisture content measurements in the top 10 cm and 30 cm.

RE:

1) We will define the soil drought conditions based on relative extractable soil water (REW) at 30cm depth during measuring period (2013-2014) in the revised manuscript. The consistent soil water threshold for sapflow will be taken in the revised manuscript. From our preliminary analysis, the plant is in drought condition when soil water content (VWC) at 30 cm depth is 0.10 $m^3$ $m^{-3}$ which is equivalent to a REW value of 0.4 as reported by Grannier et al. (Granier et al., 1999, 2003). The drought severity will be classified by REW and VWC magnitude (Granier et al., 1999, 2002; Bernier et al. 2002) in the revised manusript.

References that will be added in the revised manuscript:

- Granier et al.: Evidence for soil water control on carbon and water dynamics in European forests during the extremely dry year: 2003. Agricultural and Forest Meteorology, 143,123-145, 2007.
- Granier, A., Bre´da, N., Biron, P., Villette, S.: A lumped water balance model to evaluate duration and intensity of drought constraints in forest stands. Ecol. Model. 116, 269–283, 1999.
- Bernier, P.Y., Bre´da, N., Granier, A., Raulier, F., Mathieu, F.: Validation of a canopy gas exchange model and derivation of a soil water modifier for transpiration for sugar maple (Acer saccharum Marsh.) using sap flow density measurements. For. Ecol. Manage. 163, 185–196, 2002.

2) Information on root zone depth will be added in the methods section of revised manuscript.

Overall, more than 60% of the total roots distributed in the 0-60cm depth (Zhao et al. 2010). Our measurements showed over 80% roots distributed in the 0-60cm depth (Jia et al., 2016).

References that will be added in the revised manuscript:

- Zhao, W., Liu, B., Chang, X., Yang, Q., Yang, Y., Liu Z., Cleverly, J., Eamus, Derek.: Evapotranspiration partitioning, stomatal conductance, and components of the water balance: A special case of a desert ecosystem in China. J. Hydrol., 538, 374-386, 2016.
- Xin Jia, Tianshan Zha, Jinnan Gong, Ben Wang, et al.: Carbon and water exchange over a temperate semi-arid shrublandduring three years of contrasting precipitation and soil moisturepatterns. Agricultural and Forest Meteorology, 228, 120-129, 2016.

*If available, it will be good to include stomata closure point, wilting point, and hygroscopic point levels. Hence, the reader can judge the severity of drought. So, there will be some justification based on your drought classification. You used 0.08 (L178), 0.09 (Figure 2), and 0.11 (Figure 2). Is it 0.08 or 0.09?

RE: There are no values of stomata closure point, wilting point and hygroscopic point available at moment. We will add stomatal conductance measurements to figure 1 in the revised manuscript.

The soil drought conditions will be defined based on relative extractable soil water (REW) and VWC at 30cm depth during measuring period (2013-2014) in the revised manuscript. The drought severity will be classified by REW and VWC magnitude (Granier et al., 1999, 2002; Bernier et al. 2002). We are going to take consistent VWC threshold for both years in the revised manuscript.

*Also knowing wilting point and hygroscopic point helps us appreciating the Figure 6. You stratified soil water content based on three limits. How much severe the lowest value. My back of envelope calculation by using Campbell (1974) for sandy soil where porosity is ~0.42 (1-1.54/2.65), the wilting point (15000cm) is ~0.07. It seems your wilting point is much lower. Definitely, to appreciate the Figure 6 and drought severity, giving values are beneficial.

RE:From our preliminary analysis of REW and VWC, the plant is in drought condition when VWC at 30 cm depth is 0.10 $m^3$ $m^{-3}$ which is equivalent to a drought REW value of 0.4 reported by Granier et al. (1999, 2003). We will give the values of drought severity in the revised manuscript.

*A little more detail about vegetation setting is beneficial. LAI and plant canopy cover of shrub are beneficial. As far as I know, in Mu Us Desert dunes are migrating or semi-migrating depending on canopy cover. So, it will be beneficial for readers.

RE: We will add LAI measurements and descriptions of vegetation characteristics in the revised manuscript.

*L272-274. In your DISCUSSION, it will be good to include climate for these plant species. Because your ecosystem which is water-limited, most probably different than their study sites! For example, Huang et al. (2009) study site (L275) is in Guangxi, where annual precipitation is 1900 mm, and mean annual temperature is 19.3 °C. Most probably some/most part of the year, the ecosystem is energy-limited. So, it is not so surprising to see solar radiation control on sap flow. I could not find electronic copy of Zhang et al. (2003) work. Please include prevailing climate in their study area too.

RE: We will add prevailing climate of these species in the revised manuscript. Generally, the present result is in contrast to other shrub species. For example, it has been found that sap flow in *Haloxylon ammodendron* in northwest China, where annual precipitation is 37.9 mm, and mean annual temperature is 8.2 $^{\circ}$C, was mainly controlled by temperature (Zhang et al., 2003), while sap flow in *Cyclobalanopsis glauca* in south China, where annual precipitation is 1900 mm and mean annual temperature is 19.3 $^{\circ}$C, was controlled by both radiation and temperature with VWC not limiting (Huang et al. 2009).*"*

*L276-L278. To emphasize the importance of small events on ecological processes, I want to draw authors attention another study by Sala and Lauenroth (1982).

Sala and Lauenroth (1982) showed the ecological importance of small events (<5mm) in semiarid site where dominated by C4 grass. I will be worth to check!

Sala O.E. and W.K. Lauenroth (1982). Small rainfall events: an ecological role in semi-arid regions. Oecologia, 53 (3), 301-304.

RE: We appreciate the recommendation of this literature. We have read the paper and will add the findings in this literature as a support and generalizaion of our finding in our revised manuscript.

*Minor Points:

L69. VERB. ....low soil water availability limitS .....,

RE: We will correct word 'limit' into 'limits' in the revised manuscript.

L70. VERB. .....limitS vegetation productivity

RE: We will correct word 'limit' into 'limits' in the revised manuscript.

L73. I recommend citation for: grass replacement by shrubs.

RE: We will add the citation in the sentence "…semi-arid areas of northwestern China (Yu et al., 2004)." in the revised manuscript.

L103. Capitalization. ...the Mu Us Desert.....

RE: The name 'Mu Us desert' will be corrected into 'Mu Us Desert' in the revised manuscript.

L125. Capitalization. ...the Mu Us Desert....

RE: The name 'Mu Us desert' will be corrected into 'Mu Us Desert' in the revised manuscript.

L137. VERB. Mean height and sapwood area of sampled shrubs WERE .....

RE: We will correct it as suggested.

L156. Replace UPSILON in the equation with lower-case gamma, γ for psychrometric constant.

RE: We will be revising it as suggested.

L161. Insert a comma after "ground". ....the ground, and....

RE: We have inserted a comma in this sentence.

L180. VERB. Linear and nonlinear regression WERE .....

RE: We have revised the verb in this sentence.

L197. VERB. Total precipitation and number of rainfall events.... WERE lower than THOSE....
RE: We have revised the verb in this sentence.

L266. VERB. Synergistic interactions ..... ARE....
RE: We have revised the verb in this sentence.

L355-461. Please go through the references. Make sure the unity within the references. Journal names abbreviated some of them (L361, L367, L370 etc.), but not others (L 358, L383, L386 etc.). Choose one of them and stick with it. L424. Typo. Systems... L430. Typo. EcologY... L449. Typo. PLoS ONE. Compare with (L461 and L373). Use lower case for article names. Check (L456, L461, L416 etc.).
RE: We have carefully checked and will be revising citations and references throughout manuscript as suggested.

L541. Figure 4. I recommend following some color scheme (pattern) to represent different months such as jet etc. This change will help the readers to follow the figure easier than the current form.
RE: We have replot figure 4 using color scheme as suggested.
L553. Insert a comma after (dimensionless). .... (dimensionless), and ....
RE: We will revise it as suggested.

L554. To distinguish from straight arrows, I recommend using 'curved arrows' such as: The CURVED arrowS indicate the clockwise....
RE: We will be revising it as suggested.

L558. I recommend using 'three' instead of '3' days.
RE: The number "3" is corrected into 'three' in Fig.6 caption in the revised manuscript.

---

## Author Response (AR1)

**Response to the reviewers' comments**

Ref: doi:10.5194/bg-2016-480

Title: **Soil moisture control on sap-flow response to biophysical factors in a desert-shrub species,** *Artemisia ordosica*

Authors: TianShan Zha, Duo Qian, Xin Jia, Yujie Bai, Yun Tian , Charles P.-A. Bourque, Jingyong Ma, Wei Feng, Bin Wu, Heli Peltola

**Dear Editor,**

**Thank you very much for your and reviewers' helpful comments and suggestions for improvement of our manuscript. We have carefully looked at your comments and have revised the manuscript accordingly. Please find below our responses to your comments and/or revisions to the manuscript.**

**We look forward to your decision and the possible publication of our manuscript in the special issue of BG,** *Ecosystem processes and functioning across current and future dryness gradients in arid and semi-arid lands***.**

**Kind regards,**
**Tianshan Zha**
* * *
* * *
**Anonymous Referee #1**

General comments

*As the paper is about the soil moisture control on sap-flow and its response to meteorological variables, a physical basis for the definition for drought condition and its severity should be included. Instead, the authors keep on changing their definition of dry conditions for each year and in various figures. What is the reason for using 0.08 m3 m-3, as the threshold to identify drought periods? Is it for severe, moderate or mild drought? The analysis lacks consistency (Example figures 2 6). In section 2.4, they used 0.08 m3 m-3 as the threshold to identify drought conditions. In Figure 2, it is 0.11 $m^3$ $m^{-3}$ for the drought year 2013 and 0.09 for the wet 2014. Why don't they use 0.08 $m^3$ $m^{-3}$ in both years? They change threshold, definition of dry condition and VWC values in figure 6. I strongly suggest being consistent in their definition of drought conditions and use the same threshold in all figures.

*The root zone depth for this species is around 60 cm (Line 291). The water deep in the root zone can maintain transpiration rates even at low VWC. I think a better way is to define threshold based on root zone soil water content in this paper. Is there any field capacity or wilting point measurements available at the site? If so mention that in the paper and use relative available water content in the root zone. If not, use relative water content (based on maximum and minimum VWC values at the site) in the 30 cm soil layer to identify the drought conditions. The value of VWC shown in Figure 1 indicate that soil drying occurred mainly in shallow layer, not in the deep layer (30 cm), especially during pre and post growing periods.

RESPONSE: We agree with reviewer's suggestion of a consistent threshold of soil water content among years. The soil drought conditions were defined based on relative extractable soil water (REW) at a 30-cm depth during the measurement period (2013-2014) in the revised manuscript. The consistent soil water threshold of 0.10 $m^3 m^{-3}$ for sap flow was taken over years in the revised manuscript (Fig. 1). Interestingly, this threshold is equivalent to a drought REW value of 0.4 that was proposed by Granier et al. (1999; 2003). The plant is in drought condition, when VWC at 30 cm depth is $\leq 0.10$ $m^3$ $m^{-3}$. For details relevant to what constitutes drought condition, see the description in the revised text (see L188-195).

Literature that was added in the revised manuscript:
1. Granier et al.: Evidence for soil water control on carbon and water dynamics in European forests during the extremely dry year: 2003. Agricultural and Forest Meteorology, 143,123-145, 2007.
2. Granier, A., Bre´da, N., Biron, P., Villette, S.: A lumped water balance model to evaluate duration and intensity of drought constraints in forest stands. Ecol. Model. 116, 269–283, 1999.

*Is it possible to include transpiration (mm) values in this paper? That will add more value to understand the acclimation process of plants to the dry conditions.
RESPONSE: Yes, we have added estimated transpiration values (mm) in the revised manuscript (Fig. 2f). The transpiration has been estimated on the basis of leaf area index (LAI) and sap flow per leaf area (Equation 5, L205-208).

*The methods section reports leaf area measurements, but its values are not mentioned in the paper. Even in conclusion they have mentioned leaf expanded periods, but no data to support it.
RESPONSE: The leaf area measurements were used to calculate the sap flow rate per leaf area, which is a comparative unit with other species. We added the equation to calculate sap-flow rates per leaf area in the revised manuscript (Equation 5, L205-208). Leaf area measurements were added to revised manuscript. (L142, L152-158, Table 1)
We added observations of phenological phases in the revised manuscript (Line 135-137). Observations of phenological phases have been briefly described in section 2.1 relating to the "Experimental site", see Lines 135-137 "Normally, shrub leaf-expansion, leaf-expanded, and leaf-coloration stages begin in April, June, and September, respectively."

*Statistical significance should be evaluated for each figure and include *p*-value along with R2 in figures.
RESPONSE: We added statistical significance and R-square values in the figures of the revised manuscript.

*The 2013-2014 data shown clearly indicate that there is no direct control of VWC on sap flow (Figure 2 first vertical panel). Figure 2 also clearly shows that the relation between Js with Rs , T and VPD are non-linear (check previous comments on p-value). If the relationships are non-linear how can they explain the linear regression slopes shown in Figure 3? Is the linear relationships shown are statistically significant?
RESPONSE: We redefined VWC threshold of sap flow in the revised manuscript using pooled data over two years (Fig. 2). The drought conditions were defined by the VWC threshold and REW of 0.4 for drought conditions (Granier et al., 1999, 2002). The VWC threshold of 0.10 $m^3m^{-3}$ at our site is equivalent to a REW threshold of 0.4 (Line 188-195). Relations between mean sap-flow rate at specific times over a period of 8:00-20:00 and corresponding environmental factors from Jun. 1 to Aug. 31 period were linear ($p<0.05$; Fig. 3). Regression slopes were, therefore, used to identify the sensitivity of sap flow (degree of response) to the environmental variables (see e.g., Zha et al., 2013). (Line 212-215, Fig. 3)

RESPONSE: The reason for the delay between Js and $R_s$ could be related to available energy. Specific explanation could be that $R_s$ can force T and VPD to increase, causing a phase difference in time lags among the $J_s$-$R_s$, $J_s$-T, and $J_s$-VPD relations. Stomatal conductance gs peaks earlier than Rs. These delays reflect an acclimation of plant to dry and hot environments. We added this explanation in the revised manuscript (Line 338-343).

*Line 246-252: Figure 6 shows data from three days. What are those days in day number? A meaning full explanation should be given for the use of VWC limits. The first panel shows VWC variation within 0.001 m3 m-3! Is it meaning full considering the errors in VWC measurements? Also use only significant digits while using VWC values. The data is only three days. Is it possible to add more data in this figure, also from both years? Using only three days for this analysis is not conclusive.

RESPONSE: We added the DOY in Fig. 8 caption. This figure compared the degree of soil water control on the hysteresis between $J_s$ and $R_s$. Therefore, we took three contrasting drought classes (severe, moderate, light) to see corresponding changes in hysteresis to drought (Figure 8).

*Line 252: Figure 8 should be included in the results section.

RESPONSE: Fig.10 in the revised ms was included in the results section, as suggested (L280-282).

*Line 263-265: This is already known. Provide some references here.

RESPONSE: References were added. Similar results were reported by Qian et al., 2015 and Zha et al., 2013) (Line 295-299).

References added:
1. Qian, D., Zha, T., Jia, X., Wu, B., Zhang, Y., Bourque C. P. A., Qin, S., and Peltola, H.: Adaptive, water-conserving strategies in Hedysarum mongolicum endemic to a desert shrubland ecosystem, Environ. Earth. Sci., 74, 6039–6046, 2015.
2. Zha, T., Li, C., Kellomäki, S., Peltola, H., Wang, K.-Y., and Zhang, Y.: Controls of Evapotranspiration and CO2 Fluxes from Scots Pine by Surface Conductance and Abiotic Factors, Plos One, 8, e69027, 2013.

*Line 271: Rewrite this sentence. VWC don't show any direct effect on Js in the figure shown

RESPONSE: The sentence was rewritten as "…VWC is the most important factor modifying the response in sap flow in *Artemisia ordosica* to other environmental factors." (L304-306)

*Line 291: Provide information on root zone depth in the methods section.

RESPONSE: We added information on root zone depth in the methods section of the revised manuscript. Over 60% of the total roots were distributed in the 0-60cm depth (Zhao et al. 2010; Jia et al., 2016). (Line 143-144)

References that added:
1. Zhao, W., Liu, B., Chang, X., Yang, Q., Yang, Y., Liu Z., Cleverly, J., Eamus, Derek.: Evapotranspiration partitioning, stomatal conductance, and components of the water balance: A special case of a desert ecosystem in China. J. Hydrol., 538, 374-386, 2016.
2. Xin Jia, Tianshan Zha, Jinnan Gong, Ben Wang, et al.: Carbon and water exchange over a temperate semi-arid shrublandduring three years of contrasting precipitation and soil moisturepatterns. Agricultural and Forest Meteorology, 228, 120-129, 2016.

*Figure 1: Dotted line is not explained in the figure caption.
RESPONSE: We added an explanation in the caption of Fig. 2 in the revised ms.

*Figure 2: Use the same definition for dry periods in 2013 and 2014 as mentioned above.
RESPONSE: Consistent VWC threshold of 0.1 was used for the dry periods of 2013 and 2014 in the revised manuscript.

*Figure 8: This figure is not mentioned in results sections. Look like p value is low (both N and R2 low) and not statistically significant. If it is below 95.
RESPONSE: Fig.10 (in revised ms) was simply used to explain the hysteresis between sap flow and the environmental factors. We added some results from Fig. 10 in the result section of the revised manuscript. Statistical significance was checked. Only regression lines with p-value < 0.05 are shown in the Fig. of the revised manuscript (Fig. 10; Line 280-282).
* * *
**Anonymous Referee #2**
*General comments
I have some minor comments which should be addressed about the drought classification, soil and vegetation characteristics. Also at the end, I have some minor points about writing. Based on Li et al., 2014 (L108), shrub is shallow rooted. If available/known, it may be good to include root distribution of Artemisia ordosica such as XX% of shrub roots are located within the top 30 cm, and tap root can reach up to 60 cm (Zhao et al., 2010 from L291). This also supports your soil moisture content measurements in the top 10 cm and 30 cm.
RESPONSE:
1) We redefined VWC threshold of sap flow in the revised manuscript using pooled data over two years (Fig. 2). The drought conditions were defined by a VWC threshold and REW-value of 0.4 for drought conditions (Granier et al., 1999, 2002). The VWC threshold of 0.10 $m^3m^{-3}$ at our site is equivalent to a REW threshold of 0.4 (Line 188-195).

References added in the revised manuscript:
- Granier et al.: Evidence for soil water control on carbon and water dynamics in European forests during the extremely dry year: 2003. Agricultural and Forest Meteorology, 143,123-145, 2007.

- Granier, A., Bre´da, N., Biron, P., Villette, S.: A lumped water balance model to evaluate duration and intensity of drought constraints in forest stands. Ecol. Model. 116, 269–283, 1999.

2) Information on root zone depth was added in the methods section of the revised manuscript. Overall, more than 60% of the total roots were distributed within the 0-60 cm depth (Zhao et al. 2010; Jia et al., 2016). (L143-144)

References added in the revised manuscript:
- Zhao, W., Liu, B., Chang, X., Yang, Q., Yang, Y., Liu Z., Cleverly, J., Eamus, Derek.: Evapotranspiration partitioning, stomatal conductance, and components of the water balance: A special case of a desert ecosystem in China. J. Hydrol., 538, 374-386, 2016.
- Xin Jia, Tianshan Zha, Jinnan Gong, Ben Wang, et al.: Carbon and water exchange over a temperate semi-arid shrublandduring three years of contrasting precipitation and soil moisturepatterns. Agricultural and Forest Meteorology, 228, 120-129, 2016.

*If available, it will be good to include stomata closure point, wilting point, and hygroscopic point levels. Hence, the reader can judge the severity of drought. So, there will be some justification based on your drought classification. You used 0.08 (L178), 0.09 (Figure 2), and 0.11 (Figure 2). Is it 0.08 or 0.09?

RESPONSE: There is no value of stomata closure point, wilting point and hygroscopic point available at the moment. We added monthly means of stomatal conductance to Table 1 in the revised manuscript.

We redefined VWC threshold of sap flow in revised manuscript using pooled data over two years (Fig. 1). The drought conditions were defined by VWC threshold and REW value of 0.4 for drought condition (Granier et al., 1999, 2002). The VWC threshold of 0.10 $m^3 m^{-3}$ at our site is equivalent to a REW threshold of 0.4 (Line 188-195).

*Also knowing wilting point and hygroscopic point helps us appreciating the Figure 6. You stratified soil water content based on three limits. How much severe the lowest value. My back of envelope calculation by using Campbell (1974) for sandy soil where porosity is ∼0.42 (1-1.54/2.65), the wilting point (15000cm) is ∼0.07. It seems your wilting point is much lower. Definitely, to appreciate the Figure 6 and drought severity, giving values are beneficial.

RESPONSE: From our analysis of REW and VWC (Fig. 1, Fig. 2d,e), the plants are in drought conditions, when VWC at a 30-cm depth is $< 0.10$ $m^3$ $m^{-3}$, which is equivalent to a drought REW-value of 0.4 reported by Granier et al. (1999, 2003) (Line 188-195).

*A little more detail about vegetation setting is beneficial. LAI and plant canopy cover of shrub are beneficial. As far as I know, in Mu Us Desert dunes are migrating or semi-migrating depending on canopy cover. So, it will be beneficial for readers.

RESPONSE: We added monthly means of LAI and descriptions of vegetation characteristics in the revised manuscript (Table 1).

*L272-274. In your DISCUSSION, it will be good to include climate for these plant species. Because your ecosystem which is water-limited, most probably different than their study sites!

For example, Huang et al. (2009) study site (L275) is in Guangxi, where annual precipitation is 1900 mm, and mean annual temperature is 19.3 ℃. Most probably some/most part of the year, the ecosystem is energy-limited. So, it is not so surprising to see solar radiation control on sap flow. I could not find electronic copy of Zhang et al. (2003) work. Please include prevailing climate in their study area too.

RESPONSE: We added prevailing climate of these species in the revised manuscript. Generally, the present result is in contrast to other shrub species. For example, it has been found that sap flow in *Haloxylon ammodendron* in northwest China, where annual precipitation is 37.9 mm, and mean annual temperature is 8.2 ℃, was mainly controlled by temperature (Zhang et al., 2003), while sap flow in *Cyclobalanopsis glauca* in south China, where annual precipitation is 1900 mm and mean annual temperature is 19.3 ℃, was controlled by both radiation and temperature with VWC not limiting (Huang et al. 2009)*."* (Line 304-310)

*L276-L278. To emphasize the importance of small events on ecological processes, I want to draw authors attention another study by Sala and Lauenroth (1982).
Sala and Lauenroth (1982) showed the ecological importance of small events (<5mm) in semiarid site where dominated by C4 grass. I will be worth to check!
Sala O.E. and W.K. Lauenroth (1982). Small rainfall events: an ecological role in semi-arid regions. Oecologia, 53 (3), 301-304.

RESPONSE: We appreciate the recommendation. We have read the paper and added the relevant findings as a support and generalization of our own results in the revised manuscript (see references in the revised ms).

*Minor Points:
L69. VERB. ....low soil water availability limitS .....,
RESPONSE: We corrected this (Line 70)

L70. VERB. .....limitS vegetation productivity
RESPONSE: We corrected this as well (Line 71)

L73. I recommend citation for: grass replacement by shrubs.
RESPONSE: We added the citation in the sentence "…semi-arid areas of northwestern China (Yu et al., 2004)." in the revised manuscript (Line 74)

L103. Capitalization. ...the Mu Us Desert.....
RESPONSE: The name 'Mu Us desert' was corrected to 'Mu Us Desert' in the revised manuscript. (Line 109)

L125. Capitalization. ...the Mu Us Desert....
RESPONSE: The name 'Mu Us desert' was corrected to 'Mu Us Desert' in the revised manuscript. (Line 131)

L137. VERB. Mean height and sapwood area of sampled shrubs WERE .....
RESPONSE: We corrected this (Line 146)

L156. Replace UPSILON in the equation with lower-case gamma, γ for psychrometric constant.
RESPONSE: We revised as suggested (Line 168-170)

L161. Insert a comma after "ground". ....the ground, and....
RESPONSE: We inserted a comma in the sentence.

L180. VERB. Linear and nonlinear regression WERE .....
RESPONSE: We corrected this (Line 209)

L197. VERB. Total precipitation and number of rainfall events.... WERE lower than THOSE....
RESPONSE: Yes, we corrected this (Line 229)

L266. VERB. Synergistic interactions ..... ARE....
RESPONSE: Corrected (Line 300)

L355-461. Please go through the references. Make sure the unity within the references. Journal names abbreviated some of them (L361, L367, L370 etc.), but not others (L 358, L383, L386 etc.). Choose one of them and stick with it. L424. Typo. Systems... L430. Typo. EcologY... L449. Typo. PLoS ONE. Compare with (L461 and L373). Use lower case for article names. Check (L456, L461, L416 etc.).
RESPONSE: We carefully checked and revised citations and references throughout the manuscript as suggested.

L541. Figure 4. I recommend following some color scheme (pattern) to represent different months such as jet etc. This change will help the readers to follow the figure easier than the current form.
RESPONSE: We replotted Fig. 6 in the revised ms using color, as suggested. (Fig. 6)

L553. Insert a comma after (dimensionless). .... (dimensionless), and ....
RESPONSE: We revised the sentence.

L554. To distinguish from straight arrows, I recommend using 'curved arrows' such as: The CURVED arrowS indicate the clockwise....
RESPONSE: We revised as suggested. (Fig. 7)

L558. I recommend using 'three' instead of '3' days.
RESPONSE: We write "three" instead (Fig. 8)

**Track change manuscript version**

[revised manuscript text omitted]

---

## Author Response (AR2)

**Response to the reviewers' comments**

Ref: doi:10.5194/bg-2016-480
Title: **Soil moisture control on sap-flow response to biophysical factors in a desert-shrub species, *Artemisia ordosica***
Authors: TianShan Zha, Duo Qian, Xin Jia, Yujie Bai, Yun Tian , Charles P.-A. Bourque, Jingyong Ma, Wei Feng, Bin Wu, Heli Peltola

**Dear Editor,**

**Thank you very much for your helpful comments and suggestions for improvement of our manuscript. We have carefully looked at your comments and have revised the manuscript accordingly. Please find below our responses to your comments and/or revisions to the manuscript.**

**We look forward to your decision and the possible publication of our manuscript in the special issue of BG, *Ecosystem processes and functioning across current and future dryness gradients in arid and semi-arid lands*.**

**Kind regards,**
**Tianshan Zha**
* * *
Comments to the Author:
The manuscript was reasonably well-designed and written if not perhaps a bit standard with limited new insights into the response of sapflux to environmental variables in dryland ecosystems. The discussion was a weak point; mechanistic rigor could be added by noting important lags in the soil-vegetation-atmosphere system that must be taken into account to understand hysteresis behavior (e.g. Bohrer et al. http://onlinelibrary.wiley.com/doi/10.1029/2005WR004181/full or Matheny et al., http://onlinelibrary.wiley.com/doi/10.1002/2014JG002623/full). The discussion of hysteresis loops is interesting but must be framed more mechanistically to have full impact. I also note that the discussion of the decoupling coefficient was somewhat lacking after its introduction in the Methods section nor is it clear how u* was calculated for atmospheric conductance calculations. Please address these and the following minor comments and send a revised manuscript for consideration of publication.

RESPONSE: The discussion on hysteresis has been re-written (see line 338-356). The decoupling coefficient was used in the discussion (see line 350-351). The u* was used as input variable of equation (2) to calculate aerodynamic conductance which is used to calculate decoupling coefficient. (line 182, line 178)
* * *
Minor comments:
(All line numbers correspond to the Author's Response manuscript.)

Quantify 'sizeable' on line 455.
RESPONSE: We revised the sentence as "resulting in transpiration 34% lower in 2013 than that in 2014." (line 52-53)

'and thus less sensitive' on line 461.
RESPONSE: We revised the sentence. (line 58)

I'm not sure what the resultant for line 489 is but I assume that it will be improved by BG editorial services. (The same holds for line 509.)
RESPONSE: There is something wrong with the format. There was no 'the resultant' in clear copy of the ms.

502: usually around midday
RESPONSE: We revised the word as suggested. (line 98)

In equation 2 how was friction velocity measured? (and wind speed for that matter, which is described in the following subsection 2.3 that would be better off before the description of the decoupling coefficient.)
RESPONSE: The friction velocity was measured using eddy covariance system (line 184). We switch the order of subsection 2.2 and 2.3, so that the description of environmental measurement appears before the description of stomatal conductance and decoupling coefficient. (see line 183-184).

On line 594, why these definitions of the seasons?
RESPONSE: The reason we defined the seasons is to simply clarify the calendar period of each season (spring, summer, and autumn) that was used in the manuscript.

on line 598, these REW/VWC relationships should approximately hold for different soil types: do the references relate to the soil types found at the site? On line 604 I don't understand how this would vary from year to year.
RESPONSE: The soil types were not described, but the studies were done in forest stands with different soil types in ten European countries including Mediterranean climate, and Australia as well. Therefore, the REW of 0.4 was reasonable as a drought threshold.
The annual precipitation is different, therefore, the maximum and minimum soil water content varies with years.

Ref:
Granier, A., Bréda, N., Biron, P., and Villette, S.: A lumped water balance model to evaluate duration and intensity of drought constraints in forest stands. Ecol. Model., 116, 269–283, 1999.
Granier, A., Reichstein M., Bréda N., Janssens I. A., Falge E., Ciais P., Grünwald T.,

Aubinet M., Berbigier P., Bernhofer C., Buchmann N., Facini O., Grassi G., Heinesch B., Ilvesniemi H., Kerone P., Knohl A., Köstner B., Lagergren F., Lindroth A., Longdoz B., Loustau D., Mateus J., Montagnani L., Nys C., Moors E., Papale D., Peiffer M., Pilegaard K., Pita G., Pumpanen J., Rambal S., Rebmann C., Rodrigues A., Seufert G., Tenhunen J., Vesala T., and Wang Q.: Evidence for soil water control on carbon and water dynamics in European forests during the extremely dry year: 2003. Agr. Forest Meteorol., 143, 123-145, 2007.

Zeppel, M. J. B., Murray, B. R., Barton, C., and Eamus, D.: Seasonal responses of xylem sap velocity to VPD and solar radiation during drought in a stand of native trees in temperate Australia, Funct. Plant Biol., 31, 461-470, 2004.

Zeppel, M. J. B., Macinnis-Ng, C. M. O., Yunusa, I. A. M., Whitley, R. J., and Eamus, D. Long term trends of stand transpiration in a remnant forest during wet and dry years, J. Hydrol., 349, 200-213, 2008.

Should equation 5 be corrected for the clumpiness of shrubs on the landscape? The leaf area index applies for the area near the shrub, but I'm assuming that the space between the shrubs has marginal vegetation which should be accounted for when presenting results on a per-area basis.

RESPONSE: We calculated the leaf area index with allometric equations (see line 164-168) , which was based on measurements of sampled leaf area in the quadrat of the plot, thus clumpiness being corrected.

'Range of daily' should be 'The range of daily' in a few instances in the Results section.

RESPONSE: We revised the sentences. (line 221)

683: peaked

RESPONSE: We revised the sentence. (line 234)

wasn't an effective argument: why would you want to remove the influence of Js from the data? Perhaps its diurnal behavior to help understand lag relationships.

RESPONSE: The description in the manuscript was ambiguous, we have deleted it.

On line 766 the counterclockwise hysteresis loop is consistent with capacitance in the soil-plant-atmosphere system: it takes time for water to move up and expand vascular elements during the transition from night to day.

Please justify the statement 'In semi-arid regions, low VWC restricts plant transpiration more than VPD'. I'd assume that they are both important but perhaps at different time scales with VPD playing an important role during the mid-day depression even under high VWC, but low VWC is correlated to low VPD indicating that both water supply via VWC and demand via VPD are important in this case.

RESPONSE: The description is not clear, we have revised the discussions on the hysteresis. (see line 338-355)

I note that the decoupling coefficient was not discussed following its description in the Methods section.

RESPONSE: In the discussion, the $\Omega$ supported the result, as "This is further supported by a large $\Omega$, when VWC is high (Fig. 10b). The larger $\Omega$ is, the greater is the influence of $R_s$ on $J_s$. The effect of VWC on time lag varied between 2013 and 2014. " (line 349-350)

In Figure 6 please avoid using red and green simultaneously if possible (or at least choose different symbols so our red-green colorblind colleagues can distinguish them).

RESPONSE: We revised the figure and added different symbols.

[revised manuscript text omitted]

---

## Author Response (AR3)

**Response to the reviewers' comments**

Ref: doi:10.5194/bg-2016-480

Title: **Soil moisture control on sap-flow response to biophysical factors in a desert-shrub species,** *Artemisia ordosica*

Authors: TianShan Zha, Duo Qian, Xin Jia, Yujie Bai, Yun Tian , Charles P.-A. Bourque, Jingyong Ma, Wei Feng, Bin Wu, Heli Peltola

**Dear Editor,**

**Thank you very much for your helpful comments and suggestions in improving this manuscript. We have carefully looked at your comments and have substantially revised the manuscript accordingly. Please find below our responses to your comments and/or revisions to the manuscript.**

**We look forward to your comments and the possible publication of our manuscript in the special issue of BG,** *Ecosystem processes and functioning across current and future dryness gradients in arid and semi-arid lands***.**

**Kind regards,**
**Tianshan Zha**
* * *
Comments to the Author:

The revised manuscript by Zha et al. represents a modest improvement, but still requires substantial work. The methods and results sections are largely sound with some important avenues for further improvement noted below. I still struggle with the Introduction and Discussion (which is barely longer than three pages). Both require improvement for the manuscript to be publishable in Biogeosciences. Statements like "In general, VWC has an influence on physiological processes of plants in water-limited ecosystems (Lei et al., 2010; She et al., 2013)" do not lend confidence that this manuscript synthesizes existing knowledge effectively. The problem is that fundamental references regarding the response of leaf and canopy conductance to water availability and micrometeorological drivers is largely missing. As a consequence, the manuscript has a weak foundation and results are presented as surprising because existing knowledge has not been synthesized.

Re: We have read the relevant literature and have substantially revised the manuscript, especially the introduction and discussion. The literature information regarding the response of leaf and canopy stomatal conductance to water availability and micro-meteorological factors are added to the revised manuscript; for details see the introduction and discussion/conclusions of the revised manuscript.

At a minimum, please read (and cite if you choose):
Jarvis and McNaughton 1986:
(http://www.sciencedirect.com/science/article/pii/S0065250408601191)
Jarvis 1976: (http://rstb.royalsocietypublishing.org/content/royptb/273/927/593.full.pdf)
Oren et al. 1999:

(http://onlinelibrary.wiley.com/store/10.1046/j.1365-3040.1999.00513.x/asset/j.1365-3040.1999.00513.x.pdf?v=1&t=j2glnk7z&s=3c262f2ce665cf0835b5c17c02962ae104087e06)

Koerner 1995: (https://link.springer.com/chapter/10.1007/978-3-642-79354-7_22#page-1)

Re: We have read these papers and revised the manuscript accordingly.

I note that Jarvis and McNaughton is cited with respect to the decoupling coefficient on page 8, but this information was never synthesized in a meaningful way the Discussion.

Re: The decoupling coefficient ($\Omega$) is now discussed more fully in lines 93-95 and lines 361-363 of the manuscript.

A number of specific passages require rethinking. A selection:

"Changes in stomatal conductance and, thus, transpiration may equally affect plant water use efficiency (Pacala et al., 2001; Vilagrosa et al., 2003)" is an odd statement given that transpiration is part of the equation for water use efficiency. I recommend re-wording.

Re: We have revised those wordings throughout (e.g. see lines 95-100, or all introduction part, and discussion part).

On line 98 it need not be only mid-day.

Re: We changed this as well.

The sentence on line 105 should be cut. It seems like the authors are surprised that we know quite a bit about the controls over transpiration. The fundamental literature is largely not cited as noted. Including it seems in Artemisia species. One example: "Soil water content, in combination with other environmental factors, may have a significant influence on sap-flow rate" Of course it does when it is limiting! It's like the introduction was written with not cognizance of fundamental plant ecohydrology.

Re: We removed the sentence on line 105 and revised the introduction.

On line 187, these are the normal climatological season definitions.

Re: We removed the sentence.

"Our finding regarding lower sensitivity in Js to environmental factors (Rs, T and VPD) during dry periods was consistent with an earlier study of boreal grasslands (Zha et al., 2010)" has little meaning. How, and why was this paper written by the authors selected for a vague comparison?

Re: We have added the specific comparison to this revised version (lines 314-319), e.g. "and some other species in arid and semiarid region, e.g. sap flow in *Picea crassifolia* peaked at noon (12:00 and 14:00), and then decreased, it was heightened by increasing Rs, T, and VPD within limits (Rs < 800 W m-2, T < 18.0 ∘C and VPD < 1.4 kPa, Chang et al., 2014), and sap flow in *Caragana korshinskii* was significantly lower during the stress period, meanwhile, its conductance decreased linearly after the wilting point (She et al., 2013)."

The passage on line 247 could mean any number of things ("Soil water was shown to modify the response of Js to environmental factors (Fig. 4)." Given the specific subsequent sentences

I recommend removing it.
Re: We have revised the sentence.

On line 278, "The effect of…"
Re: We have revised the sentence.

"In general, VWC has an influence on physiological processes of plants in water-limited 302 ecosystems (Lei et al., 2010; She et al., 2013)." is not informative.
Re: We have removed the sentence.

On line 304, the literature is almost entirely cherry-picked to reflect papers of the authors rather than more relevant papers in the sapflux literature. Why are boreal grasslands and (on line 299) Scots pine chosen for comparison?
Re: We've added the relevant comparison in this revised version (see lines 314-319 in the revised manuscript).

Line 334 reiterates my point about foundational plant physiology, "According to O'Brien et al. (2004), diurnal variation in Rs could cause change in the diurnal variation in the consumption of water." Yes, of course. Plants respond to photosynthetically active radiation, which comprises the major component of incident shortwave radiation.
Re: We have removed the sentence "According to O'Brien et al. (2004), diurnal variation in Rs could cause change in the diurnal variation in the consumption of water."

The paragraph beginning on line 338 could still be written in a way that reflects that we know how stomata respond to environmental stimuli for decades or longer. Researchers have just discovered that stomatal responses to VPD are controlled by de novo synthesis of abscisic acid (McAdam et al., 2015, http://onlinelibrary.wiley.com/doi/10.1111/pce.12633/full).
Re: The paragraph was revised by focusing on acclimation to water shortage, rather than on mechanisms of hysteresis (see lines 349-363).

No meaningful discussion of the differences between 2013 and 2014 are presented in the Discussion.
Re: The results show the differences between 2013 and 2014 caused by drought, thus more drought in 2013 leading to lower gs and lower sensitivities of sap flow to micrometeorological variables (Rs, VPD, T) than in 2014. Also hysteresis between sap flow and environmental factors is larger in 2013 than those in 2014. The discussion focus on water use strategy in response to water limitation. We hope the key point of this paper in the revised manuscript could be clearer.

How certain are the values presented in Table 2 (i.e. what is the representative uncertainty)?
Re: The data are given as monthly mean (daytime) values; we discarded data on days affected by rainfall, including one day before and after rainfall and during rainfall, so that the real time lag could be determined between sap flow and the environmental variables.

**Soil moisture control on sap-flow response to biophysical factors in a desert-shrub species,** *Artemisia ordosica*

**Authors:** Tianshan Zha[1,3*#], Duo Qian[2#], Xin Jia[1,3], Yujie Bai [1], Yun Tian[1], Charles P.-A. Bourque[4], Wei Feng[1], Bin Wu[1], Heli Peltola[5]

[1.] Yanchi Research Station, School of Soil and Water Conservation, Beijing Forestry University, Beijing 100083, China

[2.] Beijing Vocational College of Agriculture, Beijing 102442, China

[3.] Key Laboratory of State Forestry Administration on Soil and Water Conservation, Beijing Forestry University, Beijing, China

[4.] Faculty of Forestry and Environmental Management, 28 Dineen Drive, PO Box 4400, University of New Brunswick, New Brunswick, E3B5A3, Canada

[5.] Faculty of Science and Forestry, School of Forest Sciences, University of Eastern Finland, Joensuu, FI-80101, Finland

[#]These authors contributed equally to this work.

**Short title:   Sap flow in** *Artemisia ordosica*

*Correspondence to*: T. Zha (tianshanzha@bjfu.edu.cn),

**Author Contribution Statement:**

Dr.'s Duo Qian and Tianshan Zha contributed equally to the design and implementation of the field experiment, data collection and analysis, and writing the first draft of the manuscript.

Dr. Xin Jia gave helpful suggestions concerning the analysis of the field data and contributed to the scientific revision and editing of the manuscript.

Prof. Bin Wu contributed to the design of the experiment.

Dr.'s Charles P.-A. Bourque and Heli Peltola contributed to the scientific revision and editing of the manuscript.

Yujie Bai, Wei Feng, and Yun Tian were involved in the implementation of the experiment and in the revision of the manuscript.

**Key Message:** This study provides a significant contribution to the understanding of acclimation processes in desert-shrub species to drought-associated stress in dryland ecosystems

**Conflict of Interest:**

This research was financially supported by grants from the National Natural Science Foundation of China (NSFC No. 31670710,No. 31670708), the National Basic Research Program of China (Grant No. 2013CB429901), and by the Academy of Finland (Project No. 14921). The project is related to the Finnish-Chinese collaborative research project, EXTREME (2013-2016), between Beijing Forestry University and the University of Eastern Finland, and USCCC. We appreciate Dr. Ben Wang, Sijing Li, Qiang Yang, and others for their help with the fieldwork. **The authors declare that they have no conflict of interest.**

**Abstract:** Current understanding of acclimation processes in desert-shrub species to drought stress in dryland ecosystems is still incomplete. In this study, we measured sap flow in *Artemisia ordosica* and associated environmental variables throughout the growing seasons of 2013 and 2014 (May-September period of each year) to better understand the environmental controls on the temporal dynamics of sap flow. We found that the occurrence of drought in the dry year of 2013 during the leaf-expansion and leaf-expanded periods caused  sap flow per leaf area ($J_s$) to decline significantly, resulting in transpiration being 34% lower in 2013 than in 2014.  Sap flow per leaf area correlated positively with radiation ($R_s$), air temperature ($T$), and water vapor pressure deficit (VPD), when volumetric soil water content (VWC) was > 0.10 $m^3 m^{-3}$. Diurnal $J_s$ was generally ahead of $R_s$ by as much as  six hours. This time lag, however, decreased with increasing VWC. Relative response of $J_s$ to the environmental variables (i.e., $R_s$, $T$, and VPD) varied with VWC, $J_s$ being more strongly controlled by plant-physiological processes during periods of dryness indicated by  a low decoupling coefficient and low sensitivity to the environmental variables . According to this study, soil moisture is shown to control sap-flow (and, therefore, plant-transpiration) response in *Artemisia ordosica* to diurnal variations in biophysical factors. This species escaped (acclimated to)  water limitation by invoking a water-conservation strategy  with the  regulation of stomatal conductance and advancement of $J_s$ peaking time, manifesting in a hysteresis effect. The findings of this study add to the knowledge of acclimation processes in desert-shrub species under drought-associated stress. This knowledge is essential  in model desert-shrub-ecosystem functioning under changing climatic conditions.

**Keywords:** sap flow; transpiration; cold-desert shrubs; environmental stress; volumetric soil water content

**1. Introduction**

Due to the low amount of precipitation and high potential evapotranspiration in desert ecosystems, low soil water availability limits both plant water- and gas-exchange and, as a consequence, limits vegetation productivity (Razzaghi et al., 2011). Shrub and semi-shrub species are replacing grass species in arid and semi-arid lands  in response to ongoing aridification of the land surface  (Huang et al., 2011a). This progression is predicted to continue under a changing climate (Houghton et al., 1999; Pacala et al., 2001; Asner et al., 2003). Studies have shown that desert shrubs are able to adapt to hot-dry environments as a result of their small plant-surface area, thick epidermal hairs, and large root-to-shoot ratios (Eberbach and Burrows, 2006; Forner et al., 2014). Plant traits related to water use are likely  to be adapt differentially with  species and habitat type (Brouillette et al., 2014). Plants may select water-acquision or water-conservaon strategies in response to water limitation (Brouillette et al., 2013). Knowledge of physiological acclimation of changing species to water shortage in deserts, particularly with respect to transpiration, is inadequate and, in the context of plant adaptation to changing climatic conditions, is a immense interest  (Jacobsen et al., 2007; Huang et al., 2011a). Transpiration maintains ecosystem balance through the soil-plant-atmosphere continuumand its magnitude and timing is related to the prevailing biophysical factors (Jarvis 1976; Jarvis and McNaughton, 1986).

Sap flow  can  be used to reflect species-specific water consumption by plant (Ewers et al., 2002; Baldocchi, 2005; Naithani et al., 2012). Sap flow can also be used  to continuously monitor canopy conductance ($g_s$) and its response to environmental variables (Ewers et al., 2007; Naithani et al., 2012). Biotic- and abiotic effects on sap flow and transpiration are often interactive and confounded. The decoupling coefficient ($\Omega$) was used to examine the relative contribution of plant control through stomatal regulation nf transpiration (Jarvis and McNaughton, 1986). with more control by stomatal regulation becomes stronger as $\Omega$ approachs zero. Stomatal conductance ($g_s$) at the plant scale exerts a  large biotic control on transpiration particularly during dry conditions (Jarvis 1976; Jarvis and McNaughton, 1986). Stomatal conductance  couples photosynthesis and transpiration (Cowan and Farquhar, 1977), making this parameter an important  component of climate models in quantifying biospheric-atmospherere interactions (Baldocchi et al., 2002).

maintains ecosystem balance through the soil-plant-atmosphere continuum, but is often affected by environment factors (Huang *et al.*, 2010; Zhao et al., 2016).

Studies have shown that xylem hydraulic conductivity was closely correlated with drought resistance (Cochard et al., 2008, 2010; Ennajeh et al., 2008). With increasing aridity, trees can were expected to show progressively lessen lower their stomatal conductance, resulting in lower transpiration (McAdam et al., 2016). Generally, desert shrubs can close their stomata, and reducereducing stomatal conductance, to reduceand reduce their water consumption by transpiration when exposed to dehydration stresses around mid-day. However,. However, but differences exist among shrub species in terms of with respect to their stomatal response to changes in soil and air and soil moisture deficits (Pacala et al., 2001).

In *Elaeagnus angustifolia*, transpiration is observed to peak at noon, i.e., just before stomatal closure at mid-day under water-deficit conditions (Liu et al., 2011), peaking earlier than radiation, temperature, and water vapor pressure deficit. This response lag or hysteresis effect have been widely noticed in dry-land species (Du et al., 2011; Naithani et al., 2012), but its function needs to be further is not completely understood. In contrast, tTranspiration in *Hedysarum scoparium* peaks multiple times during the day (Xia et al., 2007). During dry periods of the year, sap flow in *Artemisia ordosica* has been observed to be controlled by VWC at about a 30-cm depth in the soil (Li et al., 2014). For some other shrubs, sap flow has been observed to decrease rapidly when the volumetric soil water content (VWC) is lower than the water loss through evapotranspiration (Buzkova et al., 2015). On the contrary, Wwhen VWC rise after rainfall events, SsSap flow in *Caragana korshinskii* and *Hippophae rhamnoides* has been found to increase with increasing rainfall intensity (Jian et al., 2016), andbut ,sap flowthat in, whereas in *Haloxylon ammodendron*, it was found to response to precipitation variedys in response to rainfall, from an immediate decline after a heavy rainfall to no observable change after a small rainfall event (Asner et al., 2003; Zheng and Wang, 2014). Sap flow has been found to increase with increasing rainfall intensity (Jian et al., 2016). Drought-insensitive shrubs have relatively strong stomatal regulation and, therefore, tend to be insensitive to soil water deficits and rainfall, unlike their drought-sensitive counterparts (Du et al., 2011). Above all, In general, uSupportnderstandings offor the relationship between sap-flow rates in desert shrubs plants and prevailing environmental factors is decidedly variable inconsistent(McDowell et al., 2013; Sus et al., 2014), potentially varying with plant habitat and species (Liu et al., 2011). Knowledge gaps remain for desert shrubs in their responses to water shortage (McDowell et al., 2013; Sus 
[revised manuscript text omitted]

Soc. Lond. Ser. B: Biol. Sci. 273, 593–610, 1976.

Jarvis, P. G., JARVIS and McNaughton, K. G. MCNAUGHTON: Stornatal Control of Transpiration: Scaling Up from Leaf
to Region, ADVANCES IN ECOLOGICAL RESEARCHAdvances in ecological research, 15, 1-42, 1986.

Jia, X., Zha, T., Wu, B., Zhang, Y., Gong, J., Qin, S., Chen, G., Kellomäki, S., and Peltola, H.: Biophysical controls on net
ecosystem $CO^2$ exchange over a semiarid shrubland in northwest China, Biogeosciences 11, 4679-4693, 2014.

Jia, X., Zha, T, Gong, J., Wang, B., Zhang, Y., Wu, B., Qin, S., and Peltola, H.: Carbon and water exchange over a temperate
semi-arid shrubland during three years of contrasting precipitation and soil moisture patterns, Agricultural and Forest
Meteorology, 228, 120-129, 2016.

Jian, S. Q., Wu, Z. N., Hu, C. H., and Zhang, X. L.: Sap flow in response to rainfall pulses for two shrub species in the
semiarid Chinese Loess Plateau, J Hydrol Hydromech, 64, 121-132, 2016.

Larry C. Brouillette, Chase M. Mason, Rebecca Y. Shirk and Lisa A. Donovan. Adaptive differentiation of traits related to
resource use in a desert annual along a resource gradient. New Phytologist (2014) 201: 1316–1327.

Lei, H., Zhi-Shan, Z., and Xin-Rong, L.: Sap flow of Artemisia ordosica and the influence of environmental factors in a
revegetated desert area: Tengger Desert, China, Hydrological Processes, 24, 1248-1253, 2010.

Li, S. L., Werger, M. A., Zuidema, P., Yu, F., and Dong, M.: Seedlings of the semi-shrub Artemisia ordosica are resistant
to moderate wind denudation and sand burial in Mu Us sandland, China, Trees, 24, 515-521, 2010.

Li, S. J., Zha, T. S., Qin, S. G., Qian, D., and Jia, X.: Temporal patterns and environmental controls of sap flow in Artemisia
ordosica, Chinese Journal of Ecology, 33, 1-7, 2014.

Lioubimtseva, E. and Henebry, G. M.: Climate and environmental change in arid Central Asia: Impacts, vulnerability, and
adaptations, Journal of Arid Environments, 73, 963-977, 2009.

Liu, B., Zhao, W., and Jin, B.: The response of sap flow in desert shrubs to environmental variables in an arid region of
China, Ecohydrology, 4, 448-457, 2011.

Matheny, A. M., Bohrer, G., Vogel, C. S., Morin, T. H., He, L., Frasson, R. P. D. M., Mirfenderesgi, G., Schäfer, K. V, R.,
Gough, C. M., Ivanov, V. Y., and Curtis, P. S.: Species‐ specific transpiration responses to intermediate disturbance
in a northern hardwood forest, Journal of Geophysical Research: Biogeosciences, 119(12), 2292-2311,2014.

McAdam, S. A., Sussmilch, F. C. and Brodribb, T. J.: Stomatal responses to vapour pressure deficit are regulated by high
speed gene expression in angiosperms, Plant, Cell and Environment, 39, 485–491, 2016.

McDowell, N. G., et al.Fisher, R. A., Xu, C.:, 2013. Evaluating theories of drought-induced vegetationmortality using a
multimodel-experiment framework, New Phytologist, 200 (2),304–321, 2013.

Scott A. M. McAdam, Frances C. Sussmilch & Timothy J. Brodribb. Stomatal responses to vapour pressure deficit are
regulated by high speed gene expression in angiosperms. Plant, Cell and Environment (2016) 39, 485–491

Meinzer, F. C., Andrade, J. L., Goldstein, G., Holbrook, N. M., Cavelier, J., and Jackson, P.: Control of transpiration from
the upper canopy of a tropical forest: the role of stomatal, boundary layer and hydraulic architecture components, Plant,
Cell and& Environment, 20, 1242-1252, 1997.

Naithani, K. J., Ewers, B. E., and Pendall, E.: Sap flux-scaled transpiration and stomatal conductance response to soil and
atmospheric drought in a semi-arid sagebrush ecosystem, Journal of Hydrology, 464, 176-185, 2012.

O'Brien, J. J., Oberbauer, S. F., and Clark, D. B.: Whole tree xylem sap flow responses to multiple environmental variables
in a wet tropical forest, Plant, Cell & Environment, 27, 551-567, 2004.

Pacala, S. W., Hurtt, G. C., Baker, D., Peylin, P., Houghton, R. A., Birdsey, R. A., Heath, L., Sundquist, E. T., Stallard, R.
F., Ciais, P., Moorcroft, P., Caspersen, J. P., Shevliakova, E., Moore, B., Kohlmaier, G., Holland, E., Gloor, M.,
Harmon, M. E., Fan, S.-M., Sarmiento, J. L., Goodale, C. L., Schimel, D., and Field, C. B.: Consistent lLand- and
aAtmosphere-bBased U.S. cCarbon sSink eEstimates, Science, 292, 2316-2320, 2001.

Qian, D., Zha, T., Jia, X., Wu, B., Zhang, Y., Bourque, C. P., Qin, S., and Peltola, H.: Adaptive, water-conserving strategies
in Hedysarum mongolicum endemic to a desert shrubland ecosystem, Environmental Earth Sciences, 74(7), 6039,2015.

Xin-ping Wang, Benjamin Eli Schaffer, Zhenlei Yang, and Ignacio Rodriguez-Iturbe. Probabilistic model predicts dynamics of vegetation biomass in a desert ecosystem in NW China. PNAS, www.pnas.org/cgi/doi/10.1073/pnas.1703684114, 2017.

Razzaghi, F., Ahmadi, S. H., Adolf, V. I., Jensen, C. R., Jacobsen, S. E., and Andersen, M. N.: Water rRelations and tTranspiration of qQuinoa (cChenopodium quinoa wWilld.) uUnder sSalinity and sSoil dDrying, Journal of Agronomy and Crop Science, 197, 348-360, 2011.

Schwinning, S. and Sala, O. E.: Hierarchy of responses to resource pulses in arid and semi-arid ecosystems, Oecologia, 141, 211-220, 2004.

She, D., Xia, Y., Shao, M., Peng, S., and Yu, S.: Transpiration and canopy conductance of Caragana korshinskii trees in response to soil moisture in sand land of China, Agroforestry systems, 87, 667-678, 2013.

Oliver Sus,O., Rafael Poyatos, R., Josep Barba, j., Nuno Carvalhais, N., Pilar Llorens, P., Mathew Williams, M., and Jordi Martínez -
[revised manuscript text omitted]

---

## Author Response (AR4)

**Dear Editor,**

**We greatly appreciate your helpful comments and suggestions for improving this manuscript. We have carefully looked at your comments and revised the manuscript accordingly. Please find below our responses to your comments and/or revisions to the manuscript.**

**We look forward to a possible publication of the manuscript bg-2016-480 in the special issue of BG,** *Ecosystem processes and functioning across current and future dryness gradients in arid and semi-arid lands***.**

**Kind regards,**
**Tianshan Zha**
* * *
* * *
Associate Editor Decision: Publish subject to minor revisions (Editor review) (27 Jul 2017) by Paul Stoy

**Re: Thank you for your very helpful comments. We have revised the manuscript accordingly. (see a marked-up version enclosed below)**

Comments to the Author:

The manuscript represents an improvement but requires further minor edits. Comments are based on the draft with tracked edits and focus on the introduction and discussion sections.

**Re: We read the manuscript through and revised manuscript based on the editor's comments. (see a marked-up version enclosed below)**

53: Js was just defined in the previous paragraph. (see also line 61 "sap-flow" with hyphen and other abbreviation inconsistency likewise elsewhere; please be consistent with abbreviations.)

**Re: We revised all abbreviations consistently throughout text. (check in a marked-up version)**

'escaped' on line 62 is too strong a word. Every plant dreams of escaping drought limitation but arid species can't. (I do note that it is used in Briolette et al., but for reproduction timing).

**Re: The word 'escaped' was deleted. The sentence was revised accordingly like " This species acclimated to water limitations by invoking a water-conservation strategy......"( see line 63 in marked-up version)**

Line 68 can be cut because no models are involved.

**Re: The sentence referred to was cut.**

Line 91 needs re-wording

**Re: The sentence was reworded as "Transpiration maintains ecosystem balanceis controlled by stomatal through changing its conductance and pores,......". (see line 87-90 in marked-up version)**

Lines 100-105 should be moved perhaps to the methods, they are a distraction here.

**Re: The statements referred to were moved to the methods. (see line 193-196 in marked-**

**up version)**

The paragraph on line 122 is actually informative. This is effective background that educates the reader and motivates the study. (line 127 can be moved to the next paragraph to "funnel" the flow of arguments toward the importance of studying A. odorosica (i.e. undo the change on line 151). Note that it may help to write A. odorosica henceforth for brevity.

**Re: This sentence was moved to the next paragraph. (see line 132-134 in in marked-up version)**

what is 'steep' on line 345? ('steep' is qualitative).

**Re: The 'steep' was changed into 'larger'.(line 309)**

sentences like that on 357 provide nice comparisons. But after that, is Picea crassifolia a dryland species? (note bolding/lack of italics in "Caragana korshinskii".

**Re: The species Picea crassifolia was deleted. It is not really dryland species.**

The gray banding in Fig. 2 is inconsistent if it refers to VWC or REW values below the indicated thresholds (unless there is a time-integrated aspect that is best to describe in the legend).

**Re: The gray bandings in figure 2 show the long dry period with low soil moisture that is < 0.1 of VWC or 0.4 of REW.**

[revised manuscript text omitted]